

# High-resolution inverse modelling of European CH₄ emissions using novel FLEXPART-COSMO TM5 4DVAR inverse modelling system

Peter Bergamaschi[1,a], Arjo Segers[2], Dominik Brunner[3], Jean-Matthieu Haussaire[3], Stephan Henne[3], Michel Ramonet[4], Tim Arnold[5,6], Tobias Biermann[7], Huilin Chen[8], Sebastien Conil[9], Marc Delmotte[4], Grant Forster[10], Arnoud Frumau[11], Dagmar Kubistin[12], Xin Lan[13,14], Markus Leuenberger[15], Matthias Lindauer[12], Morgan Lopez[4], Giovanni Manca[1], Jennifer Müller-Williams[12], Simon O'Doherty[16], Bert Scheeren[8], Martin Steinbacher[3], Pamela Trisolino[17], Gabriela Vítková[18], Camille Yver Kwok[4]

[1]European Commission Joint Research Centre (JRC), Ispra (Va), Italy
[2]Netherlands Organisation for Applied Scientific Research (TNO), Utrecht, The Netherlands
[3]Swiss Federal Laboratories for Materials Science and Technology (Empa), Dübendorf, Switzerland
[4]Laboratoire des Sciences du Climat et de l'Environnement (LSCE-IPSL), CEA-CNRS-UVSQ, Université Paris-Saclay, 91191 Gif-sur-Yvette, France
[5]National Physical Laboratory, Teddington, UK
[6]School of GeoSciences, University of Edinburgh, Edinburgh, UK
[7]Centre for Environmental and Climate Science (CEC) Lund University, Sweden
[8]University of Groningen, Groningen, The Netherlands
[9]Agence nationale pour la gestion des déchets radioactifs (Andra), DRD/OPE, Bure, France
[10]University of East Anglia, Norwich, UK
[11]Netherlands Organisation for Applied Scientific Research (TNO), Petten, The Netherlands
[12]Deutscher Wetterdienst, Hohenpeissenberg Meteorological Observatory (MOHp), Germany
[13]Cooperative Institute for Research in Environmental Sciences, University of Colorado Boulder, Boulder, CO, USA
[14]NOAA Earth System Research Laboratory, Global Monitoring Laboratory, Boulder, CO, USA
[15]University of Bern, Physics Institute, Climate and Environmental Division, and Oeschger Centre for Climate Change Research, Bern, Switzerland
[16]Atmospheric Chemistry Research Group, University of Bristol, Bristol, UK
[17]National Research Council of Italy, Institute of Atmospheric Sciences and Climate (CNR-ISAC), Bologna, Italy
[18]Global Change Research Institute of the Czech Academy of Sciences, Brno, Czech Republic
[a]retired

*Correspondence to*: Peter Bergamaschi (peter.bergamaschi@ext.ec.europa.eu)

**Abstract.** We present a novel high-resolution inverse modelling system ("FLEXVAR") based on FLEXPART-COSMO back trajectories driven by COSMO meteorological fields at 7 km × 7 km resolution over the European COSMO-7 domain and the four-dimensional variational (4DVAR) data assimilation technique. FLEXVAR is coupled offline with the global inverse modelling system TM5-4DVAR to provide background mole fractions ("baselines") consistent with the global observations assimilated in TM5-4DVAR. We have applied the FLEXVAR system for the inverse modelling of European emissions in 2018 using 24 stations with in situ measurements, complemented with data from five stations with discrete air sampling (and additional stations outside the European COSMO-7 domain used for the global TM5-4DVAR inversions). The sensitivity of the FLEXVAR inversions to different approaches to calculate the baselines, different parameterizations of the model representation error, different settings of the prior error covariance parameters, different prior inventories and



different observation data sets are investigated in detail. Furthermore, the FLEXVAR inversions are compared to inversions

with the FLEXPART extended Kalman filter ("FLExKF") system and with TM5-4DVAR inversions at $1^{\circ} \times 1^{\circ}$ resolution over Europe. The three inverse modelling systems show overall good consistency of the major spatial patterns of the derived inversion increments and in general only relatively small differences in the derived annual total emissions of larger country regions. At the same time, the FLEXVAR inversions at $7 \text{ km} \times 7 \text{ km}$ resolution allow to better reproduce the observations than the TM5-4DVAR simulations at $1^{\circ} \times 1^{\circ}$. The three inverse models derive higher annual total $CH_4$ emissions in 2018 for

Germany, France and BENELUX compared to the sum of anthropogenic emissions reported to UNFCCC and natural emissions estimated from the Global Carbon Project $CH_4$ inventory, but the uncertainty ranges of top-down and bottom-up total emission estimates overlap for all three country regions. In contrast, the top-down estimates for the sum of emissions from the United Kingdom and Ireland agree relatively well with the total of anthropogenic and natural bottom-up inventories.

**1 Introduction**

Atmospheric methane ($CH_4$) is the second most important anthropogenic greenhouse gas (GHG) after carbon dioxide ($CO_2$) with an estimated contribution of ~16.3% (0.520 W m$^{-2}$) to the direct anthropogenic radiative forcing of all long-lived GHGs in 2020 (NOAA Annual Greenhouse Gas Index (AGGI), evaluated relative to 1750 (Butler and Montzka, 2022)). Including also additional indirect effects (e.g., production of tropospheric ozone), however, the total radiative forcing of $CH_4$ is

considerably higher, with current estimates of the emission-based effective radiate forcing (ERF) of 1.21 (0.90 to 1.51) W m$^{-2}$ (Naik et al., 2021). The current global average $CH_4$ mole fraction is 162% higher than preindustrial levels 1750 (WMO, 2021) and continues to increase with recent growth rates (2014-2020: $10.1 \pm 3.2$ ppb yr$^{-1}$) being again close to the high growth rates observed during the 1980s (1984-1989: $11.9 \pm 0.9$ ppb yr$^{-1}$), while lower growth rates were observed during the 1990s and almost zero growth rates during 2000-2006 (Dlugokencky, 2022).

Reducing $CH_4$ emissions plays an essential role to mitigate climate change, especially on the near-term (Shindell et al., 2017; Shindell et al., 2012; United Nations Environment Programme and Climate and Clean Air Coalition, 2021), due to the relatively short atmospheric lifetime of around 10 years combined with its high radiative efficiency (resulting in a global warming potential (GWP) around 80 times higher compared to $CO_2$ on a 20-year timescale (Forster et al., 2021)). The global emissions pathways to limit global warming to 1.5°C, compiled by IPCC (2018) include significant reductions of $CH_4$

emissions after 2020 (for scenarios with no or limited overshoot of temperature above the 1.5°C target). The recognition of the importance of $CH_4$ emission reductions to mitigate climate change has also led to the recent "Global Methane Pledge" (European Commission, 2021) with the collective goal to reduce methane emissions by 2030 by at least 30% compared to 2020. The development of emission reduction pathways as well as the control of international climate agreements requires the accurate quantification of current (and past) GHG emissions. For $CH_4$, however, the quantification of emissions and

sinks is particularly challenging, mainly owing to the large spatial and temporal variability of emissions from many source





sectors and consequently large uncertainties in assumed mean emission factors (e.g., for natural emissions from wetlands and anthropogenic emissions from fugitive sources such as fossil fuels (coal, oil, gas) (e.g., Brandt et al., 2014) or emissions from the waste sector). Therefore, bottom-up inventories of $CH_4$, which are compiled by scaling up emissions using activity data and emission factors, have significant uncertainties. Complementary to bottom-up inventories, inverse modelling

provides top-down emission estimates using atmospheric measurements and atmospheric transport models, by optimizing emissions from emission inventories (used as prior estimates) to get an optimal agreement between simulated and observed $CH_4$ mole fractions, taking into account the uncertainties of prior emission estimates, measurements, and model simulations (e.g., Bergamaschi et al., 2018b; Houweling et al., 2017). The Global Carbon Project $CH_4$ (GCP-$CH_4$) provides synthesis analyses of the global $CH_4$ cycle based on comprehensive sets of different bottom-up and top-down estimates from the

international science community working on this topic (Jackson et al., 2020; Saunois et al., 2020; Stavert et al., 2021). The global inverse models used in these analyses have generally relatively coarse horizontal resolution, in the range of 2.5 - 6.0º (longitude) × 1.9 - 4.0º (latitude) (Saunois et al., 2020). Therefore, such models are mainly suitable to provide information on the global and larger regional scales. In order to analyse in more detail (and more accurately) regional emissions, specific regional inversions have been performed employing regional atmospheric transport models at higher horizontal resolution

(typically in the range of ~20-100 km) and making use of the increasing number of regional in situ GHG measurements, which became available in recent years in particular in Europe and North America (e.g., Bergamaschi et al., 2018a; Ganesan et al., 2015; Lunt et al., 2021; Manning et al., 2011; Miller et al., 2013). The regional models generally require global boundary conditions, which are usually provided from global inverse models. Alternatively, global models with zooming option for the specific region of interest have been employed (Bergamaschi et al., 2018a). A specific purpose of such

regional inversions is to verify national bottom-up emission inventories reported to the United Nations Framework Convention on Climate Change (UNFCCC), finally aiming at an emission monitoring and verification system to support the international climate agreements using in situ and satellite observations (Bergamaschi et al., 2018b; Deng et al., 2021; National Research Council, 2010; Pinty et al., 2019; Pinty et al., 2017). Within the European project VERIFY (https://verify.lsce.ipsl.fr/) a pre-operational GHG verification system is currently developed, employing various state-of-

the-art global and regional atmospheric transport models (Petrescu et al., 2021a; Petrescu et al., 2021b). In order to further improve the atmospheric modelling, it is essential to further increase the spatial resolution, aiming at further improving the simulation of regional monitoring stations. A pioneering high-resolution study has been reported by Henne et al. (2016), using the FLEXPART-COSMO back trajectories driven by meteorological fields from the Swiss national weather service (MeteoSwiss) at horizontal resolution of approximately 7 km × 7 km, analysing the $CH_4$ emissions from Switzerland using

continuous measurements from 6 atmospheric monitoring stations. The authors generated the FLEXPART-COSMO sensitivity fields (based on the sampling of released particles) at even higher resolution (than the resolution of the COSMO meteorological fields) of 0.02º × 0.015º (≈ 1.7 km) over the Alpine domain (but coarser horizontal resolution of 0.16º × 0.12º (≈ 13 km) outside the Alpine domain). Since they solved the inverse problem analytically (denoted by the authors as "Bayesian method"), however, they applied a reduced grid by merging model grid cells in areas with smaller average source



sensitivities in order to reduce the size of the inversion problem to about 1000 unknowns. As alternative to the "Bayesian method", Henne et al. (2016) applied also the extended Kalman filter method described by Brunner et al. (2012), which - in contrast to the "Bayesian method" - assimilates the observations sequentially, but for computational reasons also requires the application of a reduced grid.

Aiming at high-resolution inversions of larger regions (such as the European domain), we have therefore developed a novel

inversion framework (denoted "FLEXVAR") based on the four-dimensional variational (4DVAR) data assimilation technique (Meirink et al., 2008; Talagrand and Courtier, 1987), which allows to optimize a much larger number of parameters and therefore also avoids the need to apply reduced grids. As in the Henne et al. (2016) study, the new system uses FLEXPART-COSMO back trajectories driven by COSMO meteorological fields at 7 km × 7 km resolution, but is optimizing emissions of individual grid cells over the whole European COSMO-7 domain with 393 (longitude) × 338

(latitude) grid cells. Furthermore, the new system uses background mole fractions (baselines) from global TM5-4DVAR inversions (using two different approaches), i.e., it is coupling offline the FLEXPART-COSMO inversions with the global / European TM5-4DVAR inversions.

The objective of this paper is to present the new FLEXVAR system and its application to the inversion of European $CH_4$ emissions for 2018 using a comprehensive data set from 24 stations with in situ measurements, complemented with data

from 5 stations with discrete air sampling (and additional stations outside the European COSMO-7 domain used for the global TM5-4DVAR inversions). We analyse in detail the sensitivity of the FLEXVAR inversions to internal parameterizations and model settings, as well as the sensitivity to the main model input data, i.e., prior inventories and observational data. Furthermore, we compare the FLEXVAR inversions with the extended Kalman filter ("FLExKF") method and with TM5-4DVAR inversions (at 1º × 1º resolution over Europe). Finally, we present an overall analysis of

derived European $CH_4$ emissions and comparison with emissions reported to UNFCCC for some major countries (or group of countries) which are best constrained by the available observations (Germany, France, BENELUX, and United Kingdom and Ireland).

## 2 Modelling

### 2.1 FLEXPART-COSMO back trajectories

FLEXPART is a Lagrangian atmospheric transport model that simulates the advective, turbulent and convective transport by tracking the positions of a large number of infinitesimally small air parcels, so-called particles, either forward or backward in time (Pisso et al., 2019; Stohl et al., 2005). FLEXPART is an offline model that requires meteorological fields such as 3D wind fields from a numerical weather prediction (NWP) model as input. FLEXPART-COSMO is a version of the model that is driven by the output of the NWP model COSMO, which was jointly developed by a consortium of European weather

services under the lead of the German meteorological service DWD (Baldauf et al., 2011). Different from all other FLEXPART versions, FLEXPART-COSMO operates on the native vertical grid of the driving model COSMO, which


avoids potential loss of information and inaccuracies associated with the interpolation onto a different grid. More details on the model are provided in Henne et al. (2016) and Pisso et al. (2019).

In the backward mode, particles are released at the locations of individual observations and followed backwards in time over

typically a few days. By sampling the near-surface residence times of the particles along their paths, a so-called source-receptor sensitivity matrix or "footprint" is computed, which describes the relationship between the change in mole fraction at the observation site and the fluxes discretized in space and time (Seibert and Frank, 2004). A time series of simulated mole fractions can be obtained by integrating the time series of source-receptor matrices with a discretized flux estimate. The simulations were driven by hourly output from the operational COSMO-7 analyses of the Swiss weather service MeteoSwiss

at a horizontal spatial resolution of about $7\,\text{km} \times 7\,\text{km}$, largely covering western and central Europe (Fig. 1). In the simulations used here, 50000 virtual particles were released at all observation locations every 3 hours (evenly distributed over each 3-hour time interval) and traced backwards in time for 10 days (or until individual particles left the COSMO-7 domain). Despite the high spatial resolution, the orography of the COSMO-7 model is smoothed for complex terrain, leading to differences between the model surface altitude and the altitude of the observation site. When this difference is greater than

200 m, the release height of the particles has been chosen as the average between the measurement height above sea level and the model surface altitude. This approach has been used in previous studies (Henne et al., 2016) and was found to be the most representative release height (Brunner et al., 2013).

## 2.2 Coupled FLEXPART-COSMO TM5 4DVAR inverse modelling system FLEXVAR

### 2.2.1 Inversion framework

The new coupled FLEXPART-COSMO TM5 4DVAR inverse modelling system, denoted FLEXVAR, allows to optimize emissions at grid scale using the four-dimensional variational (4DVAR) data assimilation technique (Meirink et al., 2008; Talagrand and Courtier, 1987), the FLEXPART-COSMO back trajectories described in Sect. 2.1, and background mole fractions ("baselines") from TM5-4DVAR (described in Sect. 2.2.2 and 2.4). The system follows the classical Bayesian approach minimizing the cost function $J(x)$:

$$J(x) = \tfrac{1}{2}(x - x_{\mathbf{b}})^T \mathbf{B}^{-1}(x - x_{\mathbf{b}}) + \tfrac{1}{2}(H(x) - y)^T \mathbf{R}^{-1}(H(x) - y) \tag{1}$$

where $x$ is the state vector, $x_{\mathbf{b}}$ the prior estimate of the state vector (in data assimilation usually called the "background"), $y$ the set of observations (measurements) to be assimilated, $H(x)$ the observation operator (or model operator), representing the model simulation of the observations, and $\mathbf{B}$ and $\mathbf{R}$ the error covariance matrices of the prior estimate and the observations, respectively. For the regular inversions, a semi-lognormal probability density function (pdf) is applied for the

emissions $e$ to be optimized (Bergamaschi et al., 2010), optimizing the emission deviation factors $x_{i,j,t}$:

$$e_{i,j,t}(x_{i,j,t}) = \begin{cases} e_{\mathrm{b},i,j,t} \exp(x_{i,j,t}) & \text{for } x_{i,j,t} < 0 \\ e_{\mathrm{b},i,j,t} (1 + x_{i,j,t}) & \text{for } x_{i,j,t} \geq 0 \end{cases} \tag{2}$$





for each element $e_{i,j,t}$, representing the emissions of an individual grid cell with longitude index $i$, latitude index $j$, and at emission time step $t$ (and $e_{b,i,j,t}$ the prior estimate of the emission $e_{i,j,t}$). In contrast, a linear expansion of the emission deviation factors is used (resulting in a Gaussian pdf of the emissions) for additional inversions for the evaluation of posterior uncertainties (as will be described in more detail below):

$$e_{i,j,t}(x_{i,j,t}) = e_{b,i,j,t}\left(1 + x_{i,j,t}\right) \tag{3}$$

For the prior estimate the emission deviation factors $x_{i,j,t}$ are set to zero, i.e., $e_{i,j,t}(x_{i,j,t}) = e_{b,i,j,t}$ (both for the semi-lognormal and Gaussian pdf). In this study we present inversions optimizing the monthly total emissions of all FLEXPART-COSMO grid cells (at 7 km × 7 km resolution) for one year (2018), hence the dimension of the state vector is:

$$n_{state} = n_{longitude} \times n_{latitude} \times n_{time} = 393 \times 338 \times 12 = 1594008 \tag{4}$$

The background covariance matrix $\mathbf{B}$ is parameterized as the product

$$\mathbf{B} = \mathbf{S}\,\mathbf{C}\,\mathbf{S} \tag{5}$$

where $\mathbf{S}$ is a diagonal matrix with the diagonal elements containing the uncertainty of emissions (standard deviations) and $\mathbf{C}$ is the correlation matrix, which is parametrized as a Kronecker product of a horizontal correlation matrix $\mathbf{C}_{hor}$ and a temporal correlation matrix $\mathbf{C}_t$ (as in TM5-4DVAR (Meirink et al., 2008)):

$$\mathbf{C} = \mathbf{C}_{hor} \otimes \mathbf{C}_t \tag{6}$$

The spatial covariance between two grid cells is parametrized using a Gaussian function:

$$c_{hor}(i_1, i_2, j_1, j_2) = exp\left(-\frac{1}{2}\left(\frac{d(i_1, i_2, j_1, j_2)}{L_{corr}}\right)^2\right) \tag{7}$$

where $d(i_1, i_2, j_1, j_2)$ is the distance between two grid cells (with longitude indices $i_1, i_2$ and latitude indices $j_1, j_2$), and $L_{corr}$ a pre-defined correlation length constant. The temporal correlation uses an exponential decay function:

$$c_t(t_1, t_2) = exp\left(-\frac{d_t(t_1, t_2)}{t_{corr}}\right) \tag{8}$$

where $d_t(t_1, t_2)$ is the temporal distance between two emission time steps and $t_{corr}$ a pre-defined temporal correlation scale constant.

The observation error covariance matrix $\mathbf{R}$ considers only diagonal elements (i.e., ignores any error correlation between different observations) and takes into account the uncertainties of the measurements and the model representation error:

$$\mathbf{R} = \mathbf{R}_{obs} + \mathbf{R}_{mod} \tag{9}$$

We use two different approaches to parameterize the model representation error, which are described in more detail in Sect. 2.2.3.

The observation operator $H(x)$ simulates the measurements as a function of the state vector (i.e., as function of emission deviation factors). For a given measurement $m$ in time interval $t_m$ the simulation is computed as:

$$H_m(x, t_m) = \sum_{t \in T_m} \sum_{i,j} \sum_{k \in K} e_{i,j,t}(x_{i,j,t}) / \delta h_k \cdot w_k \cdot M_{air}/M_{CH_4} \cdot 10^9 \cdot \phi_{i,j,k}^m(t, t + \delta t) \; [ppb] \tag{10}$$

using the 3-dimensional FLEXPART-COSMO footprints $\phi_{i,j,k}^m(t + \delta t)$ (units: $[1 / ((kg\ air)\ m^{-3}s^{-1})]$), described in Sect. 2.1. $M_{air}$ and $M_{CH_4}$ are the molecular masses of air and $CH_4$, respectively. $\delta h_k$ is the layer thickness of vertical layer $k$ (here we



use the two lowermost layers, each with thickness of 50 m) and $w_k$ is the weighting of layer $k$ (here 0.5 for the applied two

layers). $T_m$ represents the time interval of the applied footprints (i.e., 10 days prior to the measurements), and $\delta t$ the averaging time (3 hours) for the single footprints (computed for the time interval between $t$ and $t + \delta t$). $H_m(\boldsymbol{x}, t_m)$ represents the simulated enhancement of the CH$_4$ mole fraction above the baseline (which is evaluated by using 2 different approaches as described in Sect. 2.2.2).

The minimization of the cost function Eq. (1) requires the evaluation of the gradient of the cost function with respect to the

state vector:

$$\nabla_x J(\boldsymbol{x}) = \mathbf{B}^{-1}(\boldsymbol{x} - \boldsymbol{x_b}) + \mathbf{H}^\mathrm{T}\mathbf{R}^{-1}(H(\boldsymbol{x}) - \boldsymbol{y}) \tag{11}$$

where $\mathbf{H}^\mathrm{T}$ is the adjoint model operator, describing the sensitivity of the simulated observations with respect to changes of the state vector. $\mathbf{H}^\mathrm{T}$ can be directly computed using the FLEXPART-COSMO footprints $\phi_{i,j,k}^m (t, t + \delta t)$.

In order to achieve better convergence of the minimization algorithm, pre-conditioning is applied, transforming the state

vector $\boldsymbol{x}$ to the control vector $\boldsymbol{w}$ (similarly as e.g., in TM5-4DVAR (Meirink et al., 2008)):

$$\boldsymbol{w} = \mathbf{B}^{-1/2}(\boldsymbol{x} - \boldsymbol{x_b}) \tag{12}$$

and in reverse direction:

$$\boldsymbol{x} = \boldsymbol{x_b} + \mathbf{B}^{1/2}\boldsymbol{w} \tag{13}$$

The square root of the submatrix $\mathbf{C_t}$ is calculated by eigenvalue decomposition. However, this is not possible for the

submatrix $\mathbf{C_{hor}}$ (in contrast to TM5-4DVAR) due to the large size of this matrix of:

$$n_{\mathbf{C_{hor}}} = \left(n_{\mathrm{longitude}} \times n_{\mathrm{latitude}}\right)^2 = 132834^2 \tag{14}$$

Therefore, an Arnoldi factorization (Arnoldi, 1951) is used, computing only the largest eigenvalues/eigenvectors. Consequently, the square root matrix $\mathbf{B}^{1/2}$ and its inverse in Eq. (13) and Eq. (12) are replaced by corresponding factorizations. As default setting, we use a fraction of 1% of the eigen pairs, which is reducing the size of the control vector

to 1% of the size of the state vector. Test inversions with higher fractions of eigen pairs (up to 3%) showed that a fraction of 1% is generally sufficient for the range of correlation lengths ($L_{\mathrm{corr}}$ = 50 km ... 200 km) used in this study.

For the regular inversions, we use the limited memory quasi-Newton algorithm m1qn3 developed by Gilbert and Lemaréchal (1989), which allows the optimization also of non-linear problems (as the semi-lognormal pdf Eq. (2) introduces a non-linearity of our optimization problem). In order to evaluate the posterior uncertainties, additional inversions are performed

using a conjugate gradient algorithm (Fisher and Courtier, 1995; Lanczos, 1950; Meirink et al., 2008) for minimization and the linear expansion of the emission deviation factors (Eq. (3)). The posteriori covariance is then computed from the leading eigenvalues of the Hessian of the cost function:

$$\mathbf{B}_{\mathrm{apos}} = \mathbf{B} + \sum_{k=1}^{K}\left(\frac{1}{\theta_k} - 1\right)\left(\mathbf{B}^{1/2}v_k\right)\left(\mathbf{B}^{1/2}v_k\right)^T \tag{15}$$

with $v_k$ and $\theta_k$ the eigenvectors and eigenvalues of the Hessian of the cost function.



### 2.2.2 Baselines

Since FLEXPART-COSMO is a limited domain model, providing the source-receptor relationship due to emissions in the European COSMO-7 domain, it requires the provision of background mole fractions ("baselines"), representing the $CH_4$ mole fractions of the air masses entering the COSMO-7 domain. Here, we simulate the baselines using the global TM5-4DVAR inverse modelling system (described in Sect. 2.4), using two different approaches to couple the regional FLEXPART-COSMO inversion with TM5-4DVAR. The first approach applies the method described by Rödenbeck et al. (2009) (denoted in the following as "Rödenbeck baselines"), which is a flexible nesting scheme allowing the offline coupling of regional models with global inversions. For this purpose, we use a manipulated version of TM5 which simulates only the "cis" part of the $CH_4$ fields (representing the $CH_4$ enhancement due to emissions in the area to the FLEXPART-COSMO model transported directly to the European measurement stations, without leaving the COSMO-7 domain), denoted $\Delta c_{\text{mod 1,cis}}$. The baseline mole fractions, $c_{\text{baseline}}$, are then computed as difference of the TM5-4DVAR posterior simulation ($c_{\text{mod 1}}$) and the "cis" part (Rödenbeck et al., 2009), both sampled at the location of the corresponding station:

$$c_{\text{baseline}} = c_{\text{mod 1}} - \Delta c_{\text{mod 1,cis}} \tag{16}$$

The second approach to couple FLEXPART-COSMO with TM5-4DVAR uses the particle positions of the FLEXPART-COSMO back trajectories at the time of termination, i.e., either at the applied maximum termination time (set to 10 days in our study), or when they leave the COSMO-7 domain (which can be well before the 10 days). For each individual particle position, the $CH_4$ mole fraction is then extracted from the 3-dimensional TM5-4DVAR $CH_4$ fields. Since each 3-hourly average FLEXPART-COSMO footprint is based on the release of 50000 particles, the corresponding (3-hourly average) baseline mole fraction is computed as average of the $CH_4$ mole fraction at the termination points of all individual 50000 particles. In the following, this approach is denoted "particle position baselines".

### 2.2.3 Model representation error

We applied two different approaches to estimate the model representation error. The first approach, denoted "OBS", is similar to the method described by Henne et al. (2016), evaluating the residuals (difference between observations and model simulations) as function of $CH_4$ enhancement. However, some details of our method are different from Henne et al. (2016). Here, we use the following function to parameterize the model representation error, $\Delta y_{\text{MOD } k,i}$, as function of the absolute observed $CH_4$ enhancement (i.e., observed $CH_4$ mole fraction minus $CH_4$ background), $|y_{k,i}|$:

$$\Delta y_{\text{MOD } k,i} = f_{0,k} + \rho_k \left( |y_{k,i}| + \frac{exp(-a_k|y_{k,i}|)-1}{a_k} \right) \tag{17}$$

where $k$ is the station index, $i$ the index of the individual observational data point of the time series of station $k$, and $f_{0,k}$, $\rho_k$, and $a_k$ the three fit parameters evaluated for each station. These fit parameters are determined by calculating the fit curve of the absolute residuals, $|y_{k,i} - (\mathbf{H}_k \boldsymbol{x})_i|$, as function of $|y_{k,i}|$. For large $y_{k,i}$, the curve becomes linear with slope $\rho_k$; the minimum value of the fit function is $f_{0,k}$, and $a_k$ defines how fast the function becomes linear. The evaluation of the fit



function is generally performed iteratively. In a first step, the fit parameters are computed using the prior model simulations, while in subsequent iterations the posteriori simulations of the previous iteration are used. FLEXVAR includes an outer loop system, which allows an arbitrary number of iterations. However, tests have shown that changes after the second iteration are usually very small. Therefore, for the inversions presented in this paper, we used generally only two iterations. Figure S1
(left column) illustrates the fit functions for some selected stations.

As alternative to the model representation error "OBS" described above, a second approach, denoted "METEO", has been developed, parameterizing the model representation error as function of wind speed:

$$\Delta y_{\text{MOD } k,i} = f_{0,k} + \rho_k exp(-a_k w_{k,i})$$     (18)

where $w_{k,i}$ is the wind speed (in m/s) extracted from the COSMO model corresponding to the data point $y_{k,i}$, and $f_{0,k}$, $\rho_k$,
and $a_k$ are the 3 fit parameters for station $k$. The rationale is that if wind speed at the measurement location is low, the observed mole fraction might be more influenced by local sources of emissions and therefore less representative of the modelled mole fractions. Again, the fit parameters are determined to get an optimal fit through the absolute residuals, $\left|y_{k,i} - (\mathbf{H}_k \boldsymbol{x})_i\right|$ and they are evaluated in 2 iterations. For large wind speeds, $\Delta y_{\text{MOD } k,i}$ converges towards $f_{0,k}$, while $f_{0,k} + \rho_k$ represents the model representation error at wind speed zero. The right column of Fig. S1 shows the fit functions for the
"METEO" model representation error for some selected stations.

### 2.3 Flexpart Extended Kalman Filter inverse modelling system

FLExKF is an inverse modelling system based on an extended Kalman filter as described in detail in Brunner et al. (2012) and Brunner et al. (2017). Observations are assimilated sequentially (here day by day) to provide the best linear unbiased estimate of the emissions and their uncertainties based on measurements up to the present time of the assimilation. Rather
than estimating monthly or annual mean emissions, the system adjusts the emissions continuously as the assimilation proceeds, and in this way creates a smoothly varying emission field that later can be averaged to monthly or annual means. The filter includes a forecast step, which predicts the evolution of the emissions from one assimilation time step to the next. The simplest assumption is persistence (i.e., no change with time), but to incorporate seasonally varying a priori emissions, a non-zero forecast update was implemented that follows the linear change in a priori emissions from one month to the next.
Since the forecast step is associated with an uncertainty, the posterior uncertainty can become larger than the prior uncertainty in regions that are poorly constrained by observations. In order to avoid an unrealistic growth of the uncertainties in these regions, the posterior uncertainties are reset to the a priori uncertainty whenever they become larger.

The state vector consists of two components: (i) emissions on a reduced grid with a total of 3608 elements for inversions with observation data set O1 and 6497 elements for inversions with observation data set O2 (Sect. 3.1 and Table 1), (ii)
coefficients of an AR(1) autoregressive process describing temporal correlations in the residuals at each individual site. As described in Brunner et al. (2012), the reduced grid has high spatial resolution near the measurement sites and lower resolution further away, reflecting the reduced contribution of emissions at larger distance to observed $CH_4$ variations. It was





constructed based on the combined total annual footprint of all measurement sites. The state vector may contain the emissions directly or the logarithm of the emissions. The latter option was chosen here to enforce a positive solution, i.e.,

positive methane emissions in each grid cell.

Another option in FLExKF is to optimize baseline mole fractions at each observation site in addition to the gridded emission field. However, in the configuration used here, the baselines ("Rödenbeck baselines") described in Sect. 2.2.2 were used directly without optimization. The results of FLExKF should be readily comparable to those of FLEXVAR, since the same FLEXPART-COSMO back trajectories, baselines, and observations were used. Spatial correlations in the prior emission

uncertainties were represented in the prior error covariance matrix with a correlation length scale of 200 km (exponential decay as in Eq. (8)). The matrix was scaled such that the uncertainty of the total domain emissions was 20% (1σ).

**2.4 TM5-4DVAR inverse modelling system**

TM5-4DVAR is a global inverse modelling system based on the 4DVAR data assimilation technique, and has been described in detail by Meirink et al. (2008), while subsequent updates have been reported in Bergamaschi et al. (2018a;

2010). TM5-4DVAR uses the Eulerian atmospheric chemistry transport model TM5 (Krol et al., 2005), a two-way nested zoom model, which allows to zoom in over specific regions of interest. Here, we apply the $1° \times 1°$ zooming over the European domain $-18°... 42°$ (longitude) $\times 32°...64°$ (latitude) and an intermediate $3°$ (longitude) $\times 2°$ (latitude) zooming over the extended European domain $-30°... 48°$ (longitude) $\times 26°...70°$ (latitude), while the global domain is simulated at a horizontal resolution of $6°$ (longitude) $\times 4°$ (latitude). TM5 is an offline transport model, driven by external meteorological

data, pre-processed to provide consistent meteorological fields for the different TM5 resolutions (Krol et al., 2005). In this study, 3-hourly interpolated meteorological fields from the European Centre for Medium-Range Weather Forecasts (ECMWF) ERA-Interim reanalysis (Dee et al., 2011) have been applied, using 25 vertical layers (defined as a subset of the 60 layers of the ERA-Interim reanalysis).

As for the regular FLEXVAR inversions, a semi-lognormal pdf has been used (Eq. (2)). Minimization of the cost function

Eq. (1) is performed using the m1qn3 algorithm (Gilbert and Lemaréchal, 1989) and the adjoint of the tangent linear TM5 model (Krol et al., 2008; Meirink et al., 2008) for evaluation of the gradient of the cost function Eq. (11). Four groups of $CH_4$ emissions are optimized independently: (1) wetlands, (2) rice, (3) biomass burning, and (4) all remaining sources (Bergamaschi et al., 2018a). Uncertainties of 100% per grid-cell and month were applied for each source group with a spatial correlation length scale of 200 km (Eq. (7)). The temporal correlation time scales Eq. (8) are set to 12 months for the

"remaining" $CH_4$ sources and to zero for wetlands, rice, and biomass burning. The model representation error is parameterized as a function of local emissions and 3-dimensional gradients of simulated mole fractions (Bergamaschi et al., 2010). The photochemical sinks of $CH_4$ in the troposphere (OH), and stratosphere (OH, Cl, and O($^1$D)) are simulated as described in Bergamaschi et al. (2010). For the coupling of the FLEXPART-COSMO inversions over the European COSMO domain with the global TM5-4DVAR inversions, the baselines at the monitoring stations (within the COSMO-7 domain)

have been computed using the two different approaches described in Sect. 2.2.2.



## 3. Model input data and inversions

### 3.1 Atmospheric observations

The atmospheric observations used in this study (within the COSMO-7 domain) include ground-based $CH_4$ data for 2018 from 24 stations with in situ measurements, complemented with data from 5 stations with discrete air sampling, as compiled

in Table 1. Most of the in situ measurements (15 stations) are from the atmosphere network of the Integrated Carbon Observation System (ICOS) (Heiskanen et al., 2021), a pan-European infrastructure providing harmonized atmospheric measurements which are rigorously standardised in terms of instrumentation, calibration, air sampling and quality control, including centralised data processing and data evaluation at the ICOS Atmospheric Thematic Centre (https://icos-atc.lsce.ipsl.fr/) (Hazan et al., 2016; ICOS RI, 2020; Yver-Kwok et al., 2021). Here, we use the ICOS Atmosphere Release

of final, quality controlled data 2021-1 (ICOS RI, 2021). For station Lutjewad (LUT), the ICOS data start only on 13/08/2018. Data before that date were provided by University of Groningen, with data processing very similar to the ICOS data. Data from the ICOS station Ispra (IPR) have been further processed using the robust extraction of baseline signal (REBS) spike detection algorithm (El Yazidi et al., 2018; Ruckstuhl et al., 2012) in order to filter out data affected by nearby farming activities. In addition to the ICOS measurements, further in situ measurements have been used from the UK

Deriving Emissions linked to Climate Change (DECC) network (Bilsdale (BSD), Tacolneston (TAC), Ridge Hill (RGL), Heathfield (HFD)), from Advanced Global Atmospheric Gases Experiment (AGAGE) (Mace Head (MHD)), from University of East Anglia (Weybourne (WAO)), from Netherlands Organisation for Applied Scientific Research (TNO) (Cabauw (CBW)), from Empa (Laegern Hochwacht (LHW)), and from University of Bern (Beromünster (BRM)). The in situ measurements are complemented by discrete air samples (which are usually collected weekly) from the NOAA Earth

System Research Laboratory (ESRL) global cooperative air sampling network (NOAA, 2022), with 5 stations within the COSMO-7 domain. Additional atmospheric data used for the TM5-4DVAR inversions are compiled in Table S1 and include 6 further ICOS stations with in situ measurements and 31 NOAA discrete air sampling sites located outside the COSMO-7 domain. The selected stations outside the European $1° \times 1°$ or $3° \times 2°$ TM5 zoom regions are mostly global background stations in remote areas which can be reasonably well reproduced with the coarse global TM5 resolution of $6° \times 4°$.

The atmospheric $CH_4$ data are reported on the WMO X2004A calibration scale (Dlugokencky et al., 2005; NOAA, 2021), except the AGAGE MHD data which are reported on the Tohoku University (TU) $CH_4$ standard scale (Aoki et al., 1992; Prinn et al., 2000). Comparison of parallel measurements by NOAA and AGAGE at 5 global sites over more than 25 years showed that the two calibration scales are in close agreement, with an average ratio of 1.0002±0.0007. Therefore, no scale correction has been applied.






**Table 1: European monitoring stations used in this study. "alt" is the surface altitude (m above sea level), "s.h." is the sampling height (m) above ground, "ST" specifies the sampling type ("I": in situ measurements; "D": discrete air sample measurements).**
**The last two columns indicate the use of the corresponding station data in the observation data sets O1 and O2.**

| ID | station name | data provider | lat | lon | alt | s. h. | ST | O1 | O2 |
|---|---|---|---|---|---|---|---|---|---|
| HTM | Hyltemossa | ICOS | 56.10 | 13.42 | 115 | 150 | I | ● | ● |
| BSD | Bilsdale | DECC | 54.36 | -1.15 | 380 | 248 | I | | ● |
| LUT | Lutjewad | RUG/ICOS[1] | 53.40 | 6.35 | 1 | 60 | I | ● | ● |
| MHD | Mace Head | NOAA | 53.33 | -9.90 | 5 | 21 | D | ● | ● |
| | | AGAGE | 53.33 | -9.90 | 5 | 21 | I | | ● |
| GAT | Gartow | ICOS | 53.07 | 11.44 | 70 | 341 | I | ● | ● |
| WAO | Weybourne | UEA | 52.95 | 1.12 | 15 | 10 | I | | ● |
| TAC | Tacolneston | DECC | 52.52 | 1.14 | 56 | 100 | I | | ● |
| LIN | Lindenberg | ICOS | 52.17 | 14.12 | 73 | 98 | I | ● | ● |
| RGL | Ridge Hill | DECC | 52.00 | -2.54 | 204 | 90 | I | | ● |
| CBW | Cabauw | TNO | 51.97 | 4.93 | 0 | 200 | I | | ● |
| TOH | Torfhaus | ICOS | 51.81 | 10.53 | 801 | 147 | I | ● | ● |
| HFD | Heathfield | DECC | 50.98 | 0.23 | 150 | 100 | I | | ● |
| OXK | Ochsenkopf | NOAA | 50.03 | 11.82 | 1022 | 163 | D | ● | ● |
| KRE | Kresin u Pacova | ICOS | 49.57 | 15.08 | 534 | 250 | I | ● | ● |
| KIT | Karlsruhe | ICOS | 49.09 | 8.42 | 110 | 200 | I | ● | ● |
| SAC | Saclay | ICOS | 48.72 | 2.14 | 160 | 100 | I | ● | ● |
| OPE | Observatoire Perenne[3] | ICOS | 48.56 | 5.50 | 390 | 120 | I | ● | ● |
| TRN | Trainou | ICOS | 47.96 | 2.11 | 131 | 180 | I | ● | ● |
| HPB | Hohenpeissenberg | ICOS | 47.80 | 11.02 | 934 | 131 | I | ● | ● |
| | | NOAA | 47.80 | 11.01 | 985 | 5 | D | ● | ● |
| LHW | Laegern Hochwacht | EMPA | 47.48 | 8.40 | 840 | 32 | I | | ● |
| BRM | Beromünster | UBE | 47.19 | 8.18 | 797 | 212 | I | | ● |
| HUN | Hegyhatsal | NOAA | 46.95 | 16.65 | 248 | 96 | D | ● | ● |
| JFJ | Jungfraujoch | ICOS | 46.55 | 7.98 | 3580 | 5 | I | ● | ● |
| IPR | Ispra | ICOS/JRC[2] | 45.81 | 8.64 | 210 | 100 | I | ● | ● |
| PUY | Puy de Dome | ICOS | 45.77 | 2.97 | 1465 | 10 | I | ● | ● |
| CMN | Monte Cimone | ICOS | 44.19 | 10.70 | 2165 | 8 | I | ● | ● |
| CIB | CIBA[4] | NOAA | 41.81 | -4.93 | 845 | 5 | D | ● | ● |

[1] data since 13/08/2018 from ICOS data release; before that date from University of Groningen
[2] data filtered with REBS spike detection algorithm (see Sect. 3.1)
[3] Observatoire Perenne de l'Environnement
[4] Centro de Investigacion de la Baja Atmosfera

For the in situ measurements (which are available quasi-continuously in time) we assimilate only early afternoon data for stations in the boundary layer and night time data for mountain stations (Bergamaschi et al., 2015), selecting the 3-hour time
interval of the FLEXPART back trajectories (which are provided for [0:00 - 3:00, 3:00 - 6:00, ...] UTC) which is closest to the time interval [12:00 - 15:00] LT for the stations in the boundary layer, and [0:00 - 3:00] LT for the mountain stations,





respectively. This procedure ensures consistent averaging of the FLEXPART back trajectories and the assimilated observations over the same 3-hour time intervals. The measurement uncertainty is set to 3 ppb for all observations (for observational part $\mathbf{R}_{obs}$ of the observation error covariance matrix Eq. (9)).

In this study, we investigate two observation data sets (Table 1). The first data set, denoted O1, is considered as observational base data set and uses only the ICOS and NOAA data, while the second data set, O2, includes also all additional in situ measurements. The largest difference between the two data sets is the much better observational coverage of the British Isles in O2 with 6 in situ measurement stations located in that area, compared to only one station with discrete air sampling (MHD / NOAA) in O1.

## 3.2 Emission inventories

Three different emission inventories are used alternatively as prior estimate of the major anthropogenic $CH_4$ emissions (Table 2). The first inventory is the Emissions Database for Global Atmospheric Research (EDGAR) v6.0 (EDGAR v6.0, 2021), which provides monthly sector-specific global grid maps of emissions at horizontal resolution of $0.1° \times 0.1°$ for 2000-2018. The second inventory, TNO-VERIFYv3.0, is the third version of the TNO greenhouse gas and co-emitted species (GHGco) emission database, developed by TNO within the VERIFY project. TNO-VERIFYv3.0 provides annual European $CH_4$ emissions at a horizontal resolution of $\sim 6$ km $\times$ 6 km for the years 2005-2018, but includes monthly emission profiles. The third emission inventory has been provided by GCP-$CH_4$ (Saunois et al., 2020), globally at horizontal resolution of $1° \times 1°$ for 2000-2017. In the absence of emission data for 2018 in the GCP-$CH_4$ inventory we use here the 2017 data of this inventory. The resulting error of this 1-year inconsistency, however, is considered to be much smaller compared to the overall uncertainties of the emission inventories.

Natural $CH_4$ emissions were generally used from the GCP-$CH_4$ data set (Saunois et al., 2020), providing estimates of the climatological mean emissions of the major natural source categories. Furthermore, $CH_4$ emissions from biomass burning were taken from the Global Fire Emissions Database (GFED) version 4.1 (Van Der Werf et al., 2017). However, these were included only when using the EDGAR v6.0 or TNO-VERIFYv3.0 inventories, while the GCP-$CH_4$ (anthropogenic) data set already includes emissions from biomass burning.

Using the above emission inventories, we have assembled the emission data sets E1, E2, and E3 as compiled in Table 2 and used as prior for the different inversions described in Sect. 3.4. All emission data sets have been mapped on the COSMO-7 grid, using the Python package "emiproc" (Jähn et al., 2020), which has been integrated into the FLEXVAR inverse modelling system.






**Table 2: Emission inventories used in this study. The second column ('total') lists the total CH$_4$ emissions over the COSMO-7 domain in units of Tg CH$_4$ yr$^{-1}$ for the individual categories and the totals of each inventory. The last three columns indicate the use of the corresponding inventory data in the emission data sets E1, E2, and E3. Data for EDGARv6.0, TNO-VERIFYv3.0, and GFEDv4.1 are for 2018, GCP-CH$_4$ (anthropogenic) data are for 2017, while GCP-CH$_4$ (natural) data represent climatological mean values.**

| inventory / category | total | E1 | E2 | E3 |
|---|---|---|---|---|
| **EDGARv6.0** | | | | |
| total | 17.94 | | | |
| coal | 0.74 | ● | | |
| oil | 0.24 | ● | | |
| gas | 2.20 | ● | | |
| enteric fermentation | 6.84 | ● | | |
| manure management | 2.18 | ● | | |
| rice agriculture | 0.08 | ● | | |
| solid waste (landfills and incineration) | 3.22 | ● | | |
| wastewater treatment | 1.41 | ● | | |
| energy for buildings | 0.71 | ● | | |
| further minor anthropogenic sources | 0.33 | ● | | |
| **TNO-VERIFYv3.0** | | | | |
| total | 15.67 | | | |
| fugitive emissions | 1.86 | | ● | |
| waste | 4.29 | | ● | |
| agriculture: livestock | 8.37 | | ● | |
| agriculture: other | 0.15 | | ● | |
| other stationary combustion | 0.57 | | ● | |
| further minor anthropogenic sources | 0.44 | | ● | |
| **GCP-CH$_4$ (anthropogenic)** | | | | |
| total | 21.00 | | | |
| coal | 0.98 | | | ● |
| oil gas industry | 2.78 | | | ● |
| livestock | 9.39 | | | ● |
| agriculture: rice | 0.10 | | | ● |
| waste | 7.12 | | | ● |
| biofuels and biomass burning | 0.63 | | | ● |
| **GCP-CH$_4$ (natural)** | | | | |
| total | 2.15 | | | |
| wetlands | 1.76 | ● | ● | ● |
| geological | 0.48 | ● | ● | ● |
| termites | 0.10 | ● | ● | ● |
| ocean | 0.55 | ● | ● | ● |
| soil sink | -0.75 | ● | ● | ● |
| **GFEDv4.1** | | | | |
| biomass burning | 0.02 | ● | ● | |




## 3.3 Post-processing of gridded emission data

In order to extract from gridded emission data (on COSMO-7 grid) total emissions of countries (or group of countries), country masks have been generated using the "Natural Earth dataset" (https://www.naturalearthdata.com/), attributing each

7 km × 7 km COSMO-7 grid cell to a certain country (or sea). Offshore emissions over the sea are not included in the country totals.

Since the COSMO-7 domain does not cover the upper northern part of the UK, a correction factor of 1.057 is applied to estimate the total emissions of the country region "UK+Ireland", i.e., the "UK+Ireland" emissions extracted from the corresponding grid cells within the COSMO-7 domain are multiplied by this factor (further details see Sect. S1 of the

supplementary material). Furthermore, small correction factors are applied when extracting country total emissions from the gridded emissions of data set E3 (at horizontal resolution of $1° × 1°$), since sampling of coastal $1° × 1°$ grid cells with the corresponding 7 km × 7 km COSMO-7 grid cells leads to a loss of emissions attributed to the countries, if the emissions of the coastal $1° × 1°$ grid cell originate mainly from land (further details see Sect. S1).

## 3.4 Sensitivity inversions

Table 3 compiles the different FLEXVAR inversions presented in this paper. INV-E1-O1 represents the base inversion, using the emission data set E1 as prior, the observation data set O1, the "METEO" model representation error (Sect. 2.2.3), the "Rödenbeck baselines" (Sect. 2.2.2), and our default settings for the prior error covariance. A first set of sensitivity inversions investigates the impact of using alternatively the "particle position baselines" and the alternative parameterization "OBS" of the model representation error (and the combination of both). In a further inversion series, we analyse the

sensitivity of the inversions to the main settings of the prior error covariance matrix, i.e., for the spatial correlation length constant, $L_{corr}$, the temporal correlation scale constant, $t_{corr}$, and the assumed uncertainty of emissions per grid cell and month. Furthermore, we examine the sensitivity of the inversions to the use of the alternative emission inventories E2 and E3 as prior instead of E1, and the use of the extended observational data set O2 instead of O1.

In addition to the FLEXVAR inversions compiled in Table 3, inversions with the FLExKF system (described in Sect. 2.3)

and with TM5-4DVAR (described in Sect. 2.4) have been performed for comparison with FLEXVAR (and will be discussed in Sect. 4.3). These inversions have been made for both observational data sets, O1 and O2, using the emission inventory E3 as prior. Furthermore, additional FLExKF inversions have been performed using alternatively E1 as prior.




**Table 3: FLEXVAR sensitivity inversions. The column "prior" lists the emission data set used (Table 2) and column "obs" the observation data set (Table 1). "mre" is the applied model representation error (Sect. 2.2.3) and "baseline" lists the applied approach to calculate the baselines (Sect. 2.2.2). "$L_{corr}$" is the applied spatial correlation length constant, "$t_{corr}$" the temporal correlation scale constant, and "unc" the assumed 1-sigma uncertainty of total emissions per grid cell and month (Sect. 2.2.1).**

| inversion | prior | obs | mre | baseline | $L_{corr}$ | $t_{corr}$ | unc |
|---|---|---|---|---|---|---|---|
| INV-E1-O1 | E1 | O1 | METEO | Rödenbeck | 100 | 12 m | 100% |
| **sensitivity to baselines** | | | | | | | |
| INV-E1-O1-S1 | E1 | O1 | METEO | particle position | 100 | 12 m | 100% |
| **sensitivity to model representation error** | | | | | | | |
| INV-E1-O1-S2.1 | E1 | O1 | OBS | Rödenbeck | 100 | 12 m | 100% |
| INV-E1-O1-S2.2 | E1 | O1 | OBS | particle position | 100 | 12 m | 100% |
| **sensitivity to spatial correlation length** | | | | | | | |
| INV-E1-O1-S3.1 | E1 | O1 | METEO | Rödenbeck | 50 | 12 m | 100% |
| INV-E1-O1-S3.2 | E1 | O1 | METEO | Rödenbeck | 200 | 12 m | 100% |
| **sensitivity to prior uncertainty** | | | | | | | |
| INV-E1-O1-S4.1 | E1 | O1 | METEO | Rödenbeck | 100 | 12 m | 50% |
| INV-E1-O1-S4.2 | E1 | O1 | METEO | Rödenbeck | 100 | 12 m | 200% |
| **sensitivity to temporal correlation length** | | | | | | | |
| INV-E1-O1-S5 | E1 | O1 | METEO | Rödenbeck | 100 | 1 m | 100% |
| **sensitivity to prior inventories** | | | | | | | |
| INV-E2-O1 | E2 | O1 | METEO | Rödenbeck | 100 | 12 m | 100% |
| INV-E3-O1 | E3 | O1 | METEO | Rödenbeck | 100 | 12 m | 100% |
| **sensitivity to observations** | | | | | | | |
| INV-E1-O2 | E1 | O2 | METEO | Rödenbeck | 100 | 12 m | 100% |
| INV-E1-O2-S1 | E1 | O2 | METEO | particle position | 100 | 12 m | 100% |
| INV-E1-O2-S2.1 | E1 | O2 | OBS | Rödenbeck | 100 | 12 m | 100% |
| INV-E1-O2-S2.2 | E1 | O2 | OBS | particle position | 100 | 12 m | 100% |
| INV-E2-O2 | E2 | O2 | METEO | Rödenbeck | 100 | 12 m | 100% |
| INV-E3-O2 | E3 | O2 | METEO | Rödenbeck | 100 | 12 m | 100% |

## 4. Results and discussion

### 4.1 Sensitivity of FLEXVAR inversions to internal parameterizations and model settings

#### 4.1.1 Sensitivity of FLEXVAR inversions to baselines

Figure 1 shows maps of European $CH_4$ emissions derived for the base inversion INV-E1-O1 and the sensitivity inversion INV-E1-O1-S1, in which the "particle position baselines" were used instead of the "Rödenbeck baselines". Both inversions display in general similar spatial patterns of the inversion increments, however in most regions INV-E1-O1-S1 shows somewhat lower $CH_4$ emissions than INV-E1-O1, visible in the slightly larger areas with negative inversion increments and slightly smaller areas with positive inversion increments. Consequently, also the derived country total emissions (shown in

Fig. 5) are lower in INV-E1-O1-S1, e.g., -6.6% lower over Germany and -12.8% lower over France compared to INV-E1-O1.





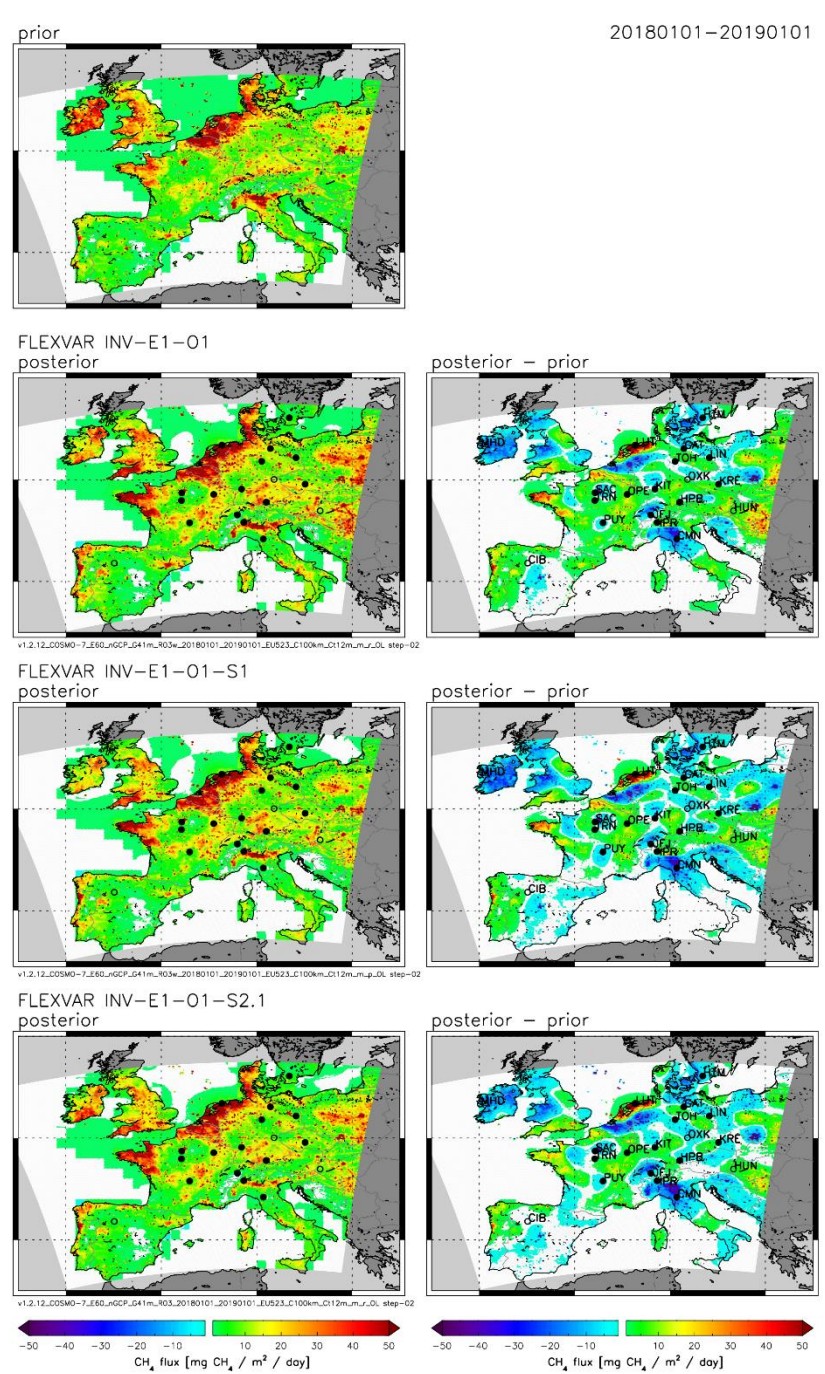

**Figure 1: Sensitivity of FLEXVAR inversions to different approaches to calculate the baselines and to parameterize the model representation error. Upper left figure: prior emissions (emission data set E1). Second row: posterior emissions (left) and difference between posterior and prior emissions (right) for base inversion INV-E1-O1 (using the Rödenbeck baselines and the METEO model representation error). Third row: inversion INV-E1-O1-S1 using the particle position baselines. Fourth row: inversion INV-E1-O1-S2.1 using the OBS model representation error. All figures show annual average CH$_4$ emissions (or change in emissions, respectively) for 2018.**



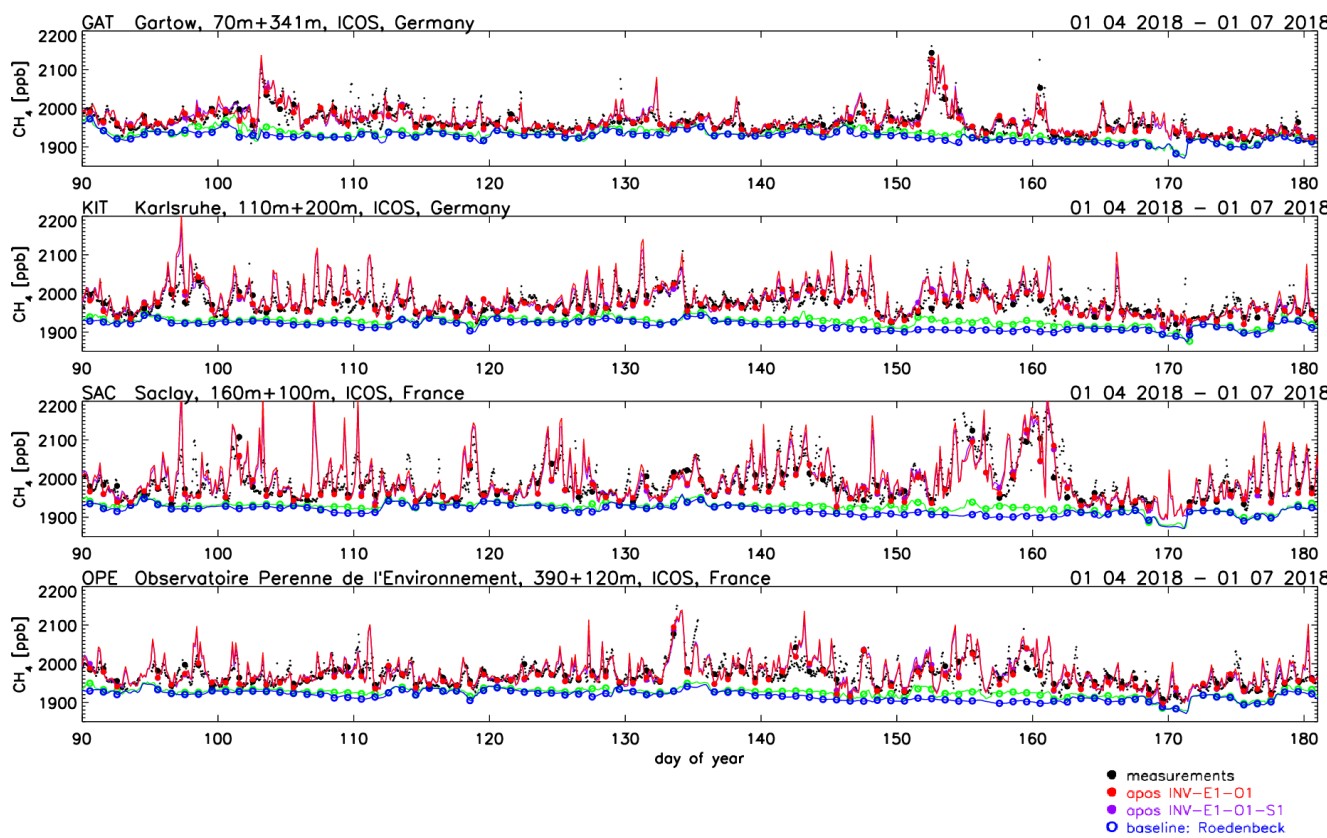

**Figure 2: Time series of simulated and observed CH₄ mole fractions at stations GAT, KIT, SAC, and OPE for 3-month period from 01 April until 01 July 2018. Blue curve shows the Rödenbeck baselines, green the particle position baselines, red the posterior CH₄ mole fractions for inversion INV-E1-O1 (using the Rödenbeck baselines), violet the posterior CH₄ mole fractions for inversion INV-E1-O1-S1 (using the particle position baselines) (owing to the similarity of both posterior simulations, however, the results of INV-E1-O1-S1 are largely overlaid by those of INV-E1-O1). Small black dots: hourly-averaged observations. Solid black circles: assimilated observations. Coloured symbols show the corresponding assimilated values (solid circles: assimilated posterior mole fractions; open circles: baseline values used for the assimilation).**

Figure 2 illustrates the two different baselines at some example stations during the 3-month period from 01 April until 01 July 2018. In general, both baselines are rather similar, including their synoptic variability. However, there are certain periods, during which the "particle position baselines" are somewhat higher than the "Rödenbeck baselines", e.g., at KIT, SAC and OPE during the period between day 140 and day 162. Consequently, the observational forcing (i.e., the enhancement of the observations above baseline) is lower during such periods for the "particle position baselines", resulting in lower derived emissions. One major difference between both approaches is that in case of the "Rödenbeck baselines" the background mole fractions are transported to the stations by TM5, while in case of the "particle position baselines" they are transported by FLEXPART. In order to further investigate which baselines are more realistic we have compared model simulations and observations for "background conditions", defined as events when the contribution of European emissions





(evaluated by Eq. (10)) is lower than a certain threshold (here set to 5 ppb). Figure S2 shows the comparison for 8 stations, for which a sufficient number (>20) of events with "background conditions" has been found. For the "Rödenbeck baselines" 6 of these 8 stations show posteriori biases close to zero (< 2 ppb), while PUY shows a small negative bias (-5.3 ppb) and

CMN a small positive bias (3.7 ppb). In contrast, the "particle position baselines" results in a smaller negative bias at PUY (-3.3 ppb), but larger positive biases at WAO (2.5 ppb), JFJ (5.5 ppb), and CMN (8.6 ppb). This analysis suggests that the performance of the "Rödenbeck baselines" is slightly better compared to the "particle position baselines" under "background conditions". However, we note that differences of the baselines shown in Fig. 2 are mainly during periods of elevated $CH_4$ enhancements, for which it is more difficult to evaluate (based on the observations) which baselines are more realistic.

**4.1.2 Sensitivity of FLEXVAR inversions to parameterization of model representation error**

Figure 1 illustrates the sensitivity of the derived emissions to the applied parameterization of the model representation error. Inversion INV-E1-O1-S2.1, for which the "OBS" model representation error has been used, results in overall lower $CH_4$ emissions compared to the base inversion INV-E1-O1 with the "METEO" parameterization, again reflected in the larger extension of the areas with negative inversion increments and smaller extension (and magnitude) of the areas with positive

inversion increments. Accordingly, the annual total emissions derived in INV-E1-O1-S2.1 are lower compared to INV-E1-O1 for all countries or group of countries (denoted in the following as "country regions") shown in Fig. 5.

The "OBS" model representation error increases with increasing observed $CH_4$ enhancement (i.e., observed $CH_4$ mole fraction minus $CH_4$ background) (Sect. 2.2.3 and Fig. S1) and shows a large dynamic range at most stations, resulting in a generally relative low weighting in the inversion of events with larger $CH_4$ enhancements. In contrast, the dynamic range of

the "METEO" model representation error is smaller at most stations, leading to a generally more equal weighting of all data points. Using the "METEO" model representation error, the observations can be better reproduced achieving a higher average correlation coefficient (r = 0.85) and lower average root mean square difference (rms = 30.0 ppb) compared to the "OBS" model representation error (r = 0.80; rms = 35.4 ppb), as shown in Fig. S3. Apart from the better statistical performance, the "METEO" model representation is probably better at estimating the capability of the model to reproduce

the observations (which largely depends on the specific meteorological situation), since wind speed might be a better indicator of the representativeness of a certain data point than the observed $CH_4$ enhancement, as the latter not only depends on the meteorological situation, but also on the regional $CH_4$ emissions.

Given the relatively large impact of the parameterization of the model representation error and the baselines, we have also performed an inversion combining the "OBS" model representation error and the "particle position baselines" (inversion

INV-E1-O1-S2.2), which yields further reduced country total emissions compared to INV-E1-O1-S2.1 and INV-E1-O1-S1 (Fig. 5).





### 4.1.3 Sensitivity of FLEXVAR inversions to model covariance settings

In the following, the sensitivity of the FLEXVAR inversions to the main parameters of the prior covariance are investigated, i.e., horizontal correlation length constant, temporal correlation scale constant, and assumed uncertainties of emissions per

grid cell and emission time step. Figure S4 shows inversions for horizontal correlation length constants $L_{\text{corr}}$ (Eq. (7)) of 50 km (INV-E1-O1-S3.1), 100 km (default value; INV-E1-O1), and 200 km (INV-E1-O1-S3.2). As expected, the spatial dimension of the inversion increments is increasing with increasing $L_{\text{corr}}$. Despite these clearly visible differences in the spatial patterns of the inversion increments, the impact on the annual total emissions of the country regions shown in Fig. 5 is relatively small, since apparently the differences in the smaller scale spatial patterns are largely averaged out over larger

areas. Associated with the increase of the horizontal correlation length constant is a significant increase of the prior uncertainties of the annual total emissions per country, since increasing horizontal correlation length constant implies larger error correlations between neighbouring grid cells and hence increasing aggregated uncertainties (as uncertainties per grid cell and month were kept constant (at 100%) in this sensitivity inversion series). Analogously, the decrease in the temporal correlation scale constant, $t_{\text{corr}}$ (Eq. (8)), results in a decrease of the aggregated annual prior uncertainty, as illustrated by

inversion INV-E1-O1-S5, in which $t_{\text{corr}}$ has been set to 1 month (instead of the default value of 12 months applied in all other inversions). Again, however, the effect on the derived annual emissions of the country regions remains very small (Fig. 5).

Figure S5 shows the dependence of the inversions on the assumed uncertainties of prior emissions per grid cell and month for values of 50% (INV-E1-O1-S4.1), 100% (default value; INV-E1-O1) and 200% (INV-E1-O1-S4.2). The increase of the

assumed prior uncertainty leads to a significant increase of the derived regional inversion increments. This effect is most pronounced at larger distances from the monitoring stations where observational constraints are relatively weak. Especially the large inversion increments visible in INV-E1-O1-S4.2 at the eastern domain boundary are probably an artefact, since the inversion may generate such patterns in regions far from the observations to compensate for systematic errors, e.g., in model transport and with little penalty in the cost function in case of assumed very high prior uncertainties.

Despite the dependence of the smaller scale regional inversion increments on the assumed prior uncertainties, the impact on the derived annual total emissions remains again very small for the country regions shown in Fig. 5, since their emissions are relatively well constrained by the available observations and since differences in the smaller scale inversion increments are averaged out over larger areas.







**Figure 3: Sensitivity of FLEXVAR inversions to applied prior emission inventories. Upper row: inversion INV-E1-O1 using emission data set E1 as prior. Middle row: inversion INV-E2-O1 using emission data set E2 as prior. Lower row: inversion INV-E3-O1 using emission data set E3 as prior. Left column: prior emissions. Middle column: posterior emissions. Right column: difference between posterior and prior emissions. All figures show annual average CH₄ emissions (or change in emissions, respectively) for 2018.**



## 4.2 Sensitivity of FLEXVAR inversions to model input data

### 4.2.1 Sensitivity of FLEXVAR inversions to prior emission inventories

Figure 3 shows maps of the European $CH_4$ emissions for INV-E1-O1, INV-E2-O1, and INV-E3-O1, which use the three different emission data sets E1, E2, and E3 (Sect. 3.2; Table 2) as prior emissions. While the major patterns of the spatial prior emission distribution look relatively similar for the three inventories (e.g., the high emissions over the BENELUX countries and the Po valley), there are significant differences in the country region total emissions (Fig. 5). E2 has lower emissions over Germany (16.1%), France (15.1%) and BENELUX (27.4%) compared to E1 (and 11.3% lower over the whole COSMO-7 domain (Table 2)), while E3 has higher total emissions over the COSMO-7 domain (15.1% higher than E1), and very high emissions especially for UK+Ireland (42.1% higher than E1). Despite these considerable differences in the prior emissions, the annual total posteriori emissions of the country regions shown in Fig. 5 are very similar for the three inversions. This indicates that the inversions are largely driven by the observations. For UK+Ireland this is somewhat surprising, since only one measurement station (MHD / NOAA) is located in this country region in the applied observation data set O1, but apparently the continental stations provide some constraints for the emissions from UK+Ireland. We will see in the next section, however, that including additional stations has a significant impact on the $CH_4$ emissions derived for UK+Ireland.

### 4.2.2 Sensitivity of FLEXVAR inversions to assimilated observations

While the base observation data set O1 uses only the ICOS in situ stations, complemented by the NOAA discrete air sampling sites, nine further in situ stations from other networks / institutions are added in observation data set O2 (Table 1). Six of the additional stations are located on the British Isles, two in Switzerland, and one in the Netherlands. Figure 4 displays the inversions INV-E1-O1 and INV-E1-O2 using the two different observation data sets. As expected, the largest differences are visible in the regions around the additional stations. For UK+Ireland, the annual total emissions are 23.0% higher in INV-E1-O2 compared to INV-E1-O1 (Fig. 5). The significant additional observational constraints for UK+Ireland are also reflected in the significantly lower posterior uncertainty for INV-E1-O2 (2-sigma uncertainty: 0.6 Tg $CH_4$ yr$^{-1}$) compared to INV-E1-O1 (2-sigma uncertainty: 1.6 Tg $CH_4$ yr$^{-1}$; Fig. 5). For the BENELUX country region only a moderate change in the annual total emissions is calculated (INV-E1-O1: 1.71 Tg $CH_4$ yr$^{-1}$; INV-E1-O2: 1.82 Tg $CH_4$ yr$^{-1}$; Fig. 5), but the spatial distribution of posteriori emissions is somewhat different, with higher emissions around the additional station CBW in INV-E1-O2 (Fig. 4). For Switzerland a larger (relative) difference of posteriori emissions is calculated, with annual total emission increasing from 0.15 Tg $CH_4$ yr$^{-1}$ (INV-E1-O1) to 0.22 Tg $CH_4$ yr$^{-1}$ (INV-E1-O2).

Using the extended observation data set O2, we have performed additional inversions, using alternatively the emission data sets E2 or E3 instead of E1. As for observation data set O1 (discussed in Sect. 4.2.1), the sensitivity of derived annual total emissions to the applied prior emission data set is relatively small (Fig. 5).







**Figure 4: Sensitivity of FLEXVAR inversions to assimilated observations. Upper left figure: prior emission. Middle row: posterior emissions (left) and difference between posterior and prior emissions (right) for inversion INV-E1-O1 (using observation data set O1). Lower row: inversion INV-E1-O2 (using observation data set O2). Solid black circles show locations of stations with in situ data, open circles locations of stations with discrete air sampling. All figures show annual average CH$_4$ emissions (or change in emissions, respectively) for 2018.**

**Figure 5: Total CH₄ emissions for Germany, France, BENELUX, and UK+Ireland derived for different sensitivity inversions (Table 3). Left: 3-month running mean total CH₄ emissions of the corresponding country regions. Right: Annual total CH₄ emissions. Open circles show prior emissions, closed circles show posterior emissions and error bars the 2-sigma uncertainties of prior and posterior emissions, respectively. The solid blue and red rectangles on the right side of the figures show the prior and posterior range from all individual inversions, and the error bars on these rectangles the minimum and maximum values of the 2-sigma uncertainties of the individual inversions.**





Furthermore, additional inversions (of observation data set O2) have been performed using alternatively the "particle
position baselines" (INV-E1-O2-S1) or the alternative parameterization "OBS" of the model representation error
(INV-E1-O2-S2.1). Similar as for observation data set O1 (discussed in Sect. 4.1.1. and 4.1.2.), the use of these alternative
parameterizations results in generally lower posteriori emissions, with lowest posteriori emission calculated in inversion
INV-E1-O2-S2.2 (combining the "OBS" model representation error and the "particle position baselines").

## 4.3 Model comparison and analysis of European CH₄ emissions

In the following we compare the FLEXVAR inversions with inversions using the extended Kalman filter ("FLExKF")
system (Sect. 2.3) and TM5-4DVAR (Sect. 2.4). Figure 6 shows the results of these three models using the emission data set
E3 as prior and the observation data set O2. Overall, all three inverse models show relatively good consistency of major
spatial patterns of the derived inversion increments, e.g., the increase of emissions over the BENELUX region and north-
western France, and the decrease of emissions around Paris compared to the prior emissions. Since FLExKF uses the same
atmospheric transport as FLEXVAR, it is to be expected that the inversions of these two models should give similar results.
Nevertheless, there are also some significant differences visible between the two models, especially for the southern part of
France, for which FLExKF yields overall lower emissions than FLEXVAR. This difference is also clearly visible in the
derived country total emissions (Fig. 7), with 10.3% lower annual total CH₄ emission for France calculated by FLExKF
(FLExKF E3-O2) compared to FLEXVAR (INV-E3-O2). In contrast, FLExKF derives somewhat higher CH₄ emissions for
BENELUX (6.3%) and UK+Ireland (6.8%) than FLEXVAR, while emissions derived for Germany are very similar (within
1.4%). One major difference between FLExKF and FLEXVAR is the different parameterization of the model representation
error, leading to a different weighting of the individual observational data points, which can cause differences in the
calculated regional inversion increments as shown for FLEXVAR in Section 4.1.2. Another difference is the magnitude of
the prior uncertainties, though this was shown for FLEXVAR to have a rather small impact on total emissions for the
country regions presented in Fig. 5. Furthermore, it is likely that the different inversion techniques have some impact on the
calculated solutions. For example, FLExKF yields generally smoother seasonal variations of derived emissions, while
FLEXVAR shows larger month-to-month variability. The latter are, however, largely filtered out by the use of 3-month
running mean values for the seasonal variation of the total emissions of country regions shown in Fig. 7 (left column).
The spatial emission patterns derived by TM5-4DVAR are in general similar to those calculated by FLEXVAR and FLExKF
(Fig. 6), but show also some differences, e.g., around the stations PUY and HPB, where TM5-4DVAR calculates higher
emissions than FLEXVAR and FLExKF, probably related to the particular challenge to simulate mountain sites and sites in
complex topography. Further differences between the models are the different derived seasonal variations of emissions, with
larger variations calculated by TM5-4DVAR for Germany, France, and UK+Ireland compared to FLEXVAR and FLExKF
(while the FLEXVAR inversions using the observation data set O2 show larger variations for BENELUX than the other
models). In addition to the different model representation error in TM5-4DVAR, very likely the fundamentally different
nature of the models (Eulerian vs. Lagrangian) and the related different simulation of transport plays an essential role.





**Figure 6: Annual average CH₄ emissions derived for year 2018 using FLEXVAR (upper row), FLExKF (middle row), and TM5-4DVAR (lower row). Left column: prior emissions. Middle column: posterior emissions. Right column: difference between posterior and prior emissions. All three inversions shown here use the same inventory data set E3 as prior and the observation data set O2.**

**Figure 7: Total CH₄ emissions for Germany, France, BENELUX, and UK+Ireland derived by the three different inverse modelling systems FLEXVAR, FLExKF, and TM5-4DVAR. For FLEXVAR only a subset of inversions is displayed here, while the whole range from all FLEXVAR sensitivity inversions is shown by the first pair (from left to right) of solid rectangles which is identical to the pair of rectangles shown in Fig. 5. The second pair of rectangles shows the range of prior (blue) and posterior (red) CH₄ emissions from all three models (and the error bars the minimum and maximum values of the 2-sigma uncertainties of all individual inversions). The black symbols show the anthropogenic CH₄ emissions reported to UNFCCC (and their estimated 2-sigma uncertainties), blue symbols the natural emissions estimated from the GCP CH₄ inventory, and the violet symbols the sum of anthropogenic and natural bottom-up inventories.**





Nevertheless, the differences in the annual total emissions for the country regions are only moderate. For Germany, somewhat higher emissions are calculated by TM5-4DVAR compared to FLEXVAR and FLExKF, while the posterior emissions for France, BENELUX, and UK+Ireland derived by TM5-4DVAR are in the range of emissions calculated by
FLEXVAR and FLExKF.

Figure 7 also includes inversions of the three models using the base observation data set O1. As discussed for FLEXVAR in Sect. 4.2.2., also FLExKF and TM5-4DVAR show higher emissions for UK+Ireland, when using O2 instead of O1 due to the 6 additional stations in data set O2 in that area. Furthermore, FLExKF inversions have also been performed using E1 instead of E3 as prior emissions (Fig. 7). As for FLEXVAR (Sect. 4.2.1), the impact on derived emissions is only relatively
small.

In order to evaluate the quality of the derived emissions it is useful to analyse how well the observations are reproduced by the models. Figure S6 compares the statistics (correlation coefficient and rms difference) for the three models (using emission data set E3 and observation data set O2). At most stations relatively high correlation coefficients and low rms differences are obtained by all three models. However, stations with larger regional emissions (e.g., LUT, CBW, BRM, IPR)
or complex topography (e.g., OXK, IPR) show generally poorer statistical performance. Figure S6 also shows that the best statistical performance is achieved by FLEXVAR with a mean correlation coefficient of r=0.86 (FLExKF: r=0.84, TM5-4DVAR: r=0.81) and a mean rms difference of 28.21 ppb (FLExKF: 30.53 ppb, TM5-4DVAR: 31.82 ppb). This finding demonstrates that the high spatial resolution of FLEXVAR and FLExKF at 7 km × 7 km allows to somewhat better reproduce the observations than the TM5-4DVAR simulations at 1° × 1°, although - beside the different spatial resolution -
also other factors (such as fundamental differences in the modelling of transport) are likely to play a role. The slightly better statistical performance of FLEXVAR compared to FLExKF could be due to the higher degree of freedom to optimize the emissions in FLEXVAR, but may also be partly related to other factors, such as different weighting of observations due to different parameterizations of the model representation error.

Figure S7 shows the time series of observed and simulated CH$_4$ mole fractions for all stations (inversion INV-E1-O2),
illustrating that in general the synoptic variability is well reproduced at most sites. Furthermore, FLEXVAR also simulates the average diurnal cycle at most sites realistically.

In the following, we compare the annual total CH$_4$ emissions derived by the inverse models with the anthropogenic CH$_4$ emissions reported by the countries to UNFCCC (UNFCCC, 2021). For a consistent comparison, it is necessary to take into account also estimates of the natural CH$_4$ emissions, for which we use the bottom-up inventories of natural sources from the
GCP-CH$_4$ data set (Saunois et al., 2020) (Table 2). Furthermore, the comparison of top-down and bottom-up emission estimates requires to include estimates of their uncertainties. For the uncertainty estimate of the inverse models, we use the range of results from the individual inversions (shown by the red solid rectangles in Fig. 7) and the minimum-maximum values of the 2-sigma uncertainty ranges based on the uncertainties computed for the individual inversions (shown by the error bars). The total uncertainty ranges are evaluated separately (1) for the whole set of FLEXVAR sensitivity inversions
(as shown in Fig. 5) and (2) for the whole set of all inversions, i.e., including also all FLExKF and TM5-4DVAR inversions.



The uncertainties of the UNFCCC emissions are based on the uncertainties reported by the countries for the major $CH_4$ source categories, while estimates of the uncertainty of total $CH_4$ emissions are not provided by the countries. As in Bergamaschi et al. (2015), we estimate the total uncertainties from the reported uncertainties per category, assuming - among other things - uncorrelated uncertainties for the different major source categories (for further details see section S2 in the supplementary material). The uncertainties of natural $CH_4$ emissions from wetlands were estimated from the ensemble of wetland models used for the GCP-$CH_4$ wetland emissions, taking the minimum-maximum range of the 11 individual wetland models (Poulter et al., 2017). For other natural $CH_4$ emissions, we assume an uncertainty of 100%.

Figure 7 shows that the $CH_4$ emissions estimated by the inverse models are higher than the sum of anthropogenic (UNFCCC) and natural bottom-up inventories for Germany, France and BENELUX, but the uncertainty ranges of top-down and bottom-up estimates overlap for all three country regions. The smallest overlap, however, is found for BENELUX. In contrast, the top-down estimates for UK+Ireland agree relatively well with the total of anthropogenic and natural bottom-up inventories. A tendency to higher top-down emissions compared to the total (anthropogenic and natural) bottom-up inventories for Germany, France and BENELUX has also been found in the analysis reported by Bergamaschi et al. (2018a) for the period 2006-2012, but also in that study uncertainty ranges of bottom-up and top-down estimates were overlapping. Similar tendencies to higher top-down emissions are apparent in the VERIFY analyses for the period 2005-2017 (VERIFY, 2021) using a larger ensemble of regional inversions, while global inversions (with coarser resolution) showed in general lower emissions, closer to the UNFCCC estimates for these country regions. Based on the observation that several models showed clear seasonal cycles of the derived emissions with maximum during summer, Bergamaschi et al. (2018a) suggested that higher natural emissions could explain the difference between top-down and bottom-up estimates. The FLEXVAR and FLExKF inversions analysed in this study, however, show in general only relatively small seasonal variations for Germany, France and UK+Ireland compared to TM5-4DVAR. The use of seasonal cycles to disentangle anthropogenic and natural sources is further hampered by the fact that the seasonal cycles of major anthropogenic sources are still not well characterized. Also, the anthropogenic emission inventories used in this study show rather different seasonal cycles. Most of the anthropogenic GCP-$CH_4$ emission categories (which are largely based on EDGARv4.3.2, except biomass burning (Saunois et al., 2020)) have no seasonality, except emissions from rice agriculture and biomass burning, which however play only a minor role in Europe. EDGARv6.0 (used for E1) has small seasonal variations of most energy related source categories, but assumes constant emissions for the agricultural sources (except rice) and for waste emission. In contrast, most sectors of the TNO-VERIFYv3.0 inventory (used for E2) show seasonal variations, including significant seasonal variations of all agricultural sources, resulting in significant seasonal variations of the total anthropogenic emissions with maximum emissions in September (Fig. 5).



## 5. Conclusions

We have presented the novel inverse modelling system FLEXVAR based on the 4DVAR assimilation technique and FLEXPART-COSMO back trajectories driven by COSMO meteorological fields at $7\,\text{km} \times 7\,\text{km}$ resolution over the European COSMO-7 domain. A major advantage of the 4DVAR technique is that it allows to constrain a much larger number of variables (in our study about 1.6 million) compared to analytical inversion techniques. The offline coupling with TM5-4DVAR ensures that the background mole fractions ("baselines") used in FLEXVAR are consistent with the global observations assimilated in TM5-4DVAR. We have applied the FLEXVAR system for inversions of European CH$_4$ emissions 2018 using 24 stations with in situ measurements, complemented with data from 5 stations with discrete air sampling (and additional stations outside the European COSMO-7 domain used for the global TM5-4DVAR inversions).

We have investigated the sensitivity of the FLEXVAR inversions to internal parameterizations, model settings, and main model input data. Using the "particle position baselines" yields in general lower derived emissions compared to inversions which apply the "Rödenbeck baselines", resulting in differences in the annual total emissions of 5 - 14 % for the analysed country regions (Germany, France, BENELUX, UK+Ireland). Furthermore we found a significant impact of the applied parameterization of the model representation error. Inversions using the "OBS" model representation error derive over large parts of the domain somewhat lower emissions compared to the "METEO" model representation error, with differences in the annual total emissions of 0 - 15 % for the analysed country regions. Varying the main parameters of the prior covariance (i.e., horizontal correlation length constant, temporal correlation scale constant, and assumed uncertainties of emissions per grid cell and month) has clearly visible effects on the smaller scale regional inversion increments, but the impact on the derived annual total emissions remains very small for the analysed country regions, since the differences in the smaller scale spatial patterns are largely averaged out over larger areas. Furthermore, the dependence of derived emissions on the applied prior emission inventory has been found to be relatively small for the country regions which are well constrained by the observations. Changing these observational constraints by including additional sites, however, has a significant impact on the inversions especially in the surroundings of these sites. Using the extended observation data set O2 (which includes 6 additional in situ stations located on the British Isles) yields 23 - 28 % higher emissions for UK+Ireland compared to inversions using only the base observation data set O1. At the same time, the calculated uncertainty of the posteriori emissions for UK+Ireland is significantly reduced by these additional observational constraints.

The comparison of the FLEXVAR inversions with inversions using the extended Kalman filter ("FLExKF") system (which both use the same atmospheric transport model) shows overall good consistency of major spatial patterns of the derived inversion increments, but some difference (7-10%) for the derived total CH$_4$ emission of France, probably mostly related to the use of different parameterizations of the model representation error. TM5-4DVAR also shows in general similar inversion increments and derives posterior emissions for France, BENELUX, and UK+Ireland in the range of emissions calculated by FLEXVAR and FLExKF. For Germany, however, TM5-4DVAR estimates 5-11% higher emissions than the other two models.





The FLEXVAR and FLExKF inversions at high spatial resolution of 7 km × 7 km allow to better reproduce the observations

than the TM5-4DVAR simulations at 1° × 1°, reflected in the achieved higher correlation coefficient and lower rms difference between simulations and observations. Furthermore, the statistical performance of FLEXVAR is slightly better than that of FLExKF, which could be due to the higher degree of freedom to optimize the emissions in FLEXVAR, but could be partly related also to other differences of the inversions, as e.g., the different parameterizations of the model representation error.

The inverse models derive higher annual total $CH_4$ emissions in 2018 for Germany, France and BENELUX compared to the sum of emissions reported to UNFCCC and natural emissions (estimated from the GCP-$CH_4$ inventory), but the uncertainty ranges of top-down and bottom-up estimates overlap for all three country regions. In contrast, the top-down estimates for UK+Ireland agree relatively well with the total of anthropogenic and natural bottom-up inventories.

**Code and data availability**

The code of the FLEXVAR inverse modelling system is available upon request. The atmospheric observations from ICOS are available at: https://www.icos-cp.eu/data-products/atmosphere-release. NOAA data are available at: https://gml.noaa.gov/aftp/data/greenhouse_gases/ch4/flask/, AGAGE data at: https://agage2.eas.gatech.edu/data_archive/agage/, UK DECC data at https://archive.ceda.ac.uk/

**Author contribution**

PB led the FLEXVAR development. AS designed the FLEXVAR concept and performed the technical implementation of the FLEXVAR code. PB performed the FLEXVAR and TM5-4DVAR inversions. DB performed the FLExKF inversions and contributed to the integration of the FLEXPART-COSMO back trajectories into FLEXVAR. JMH generated the FLEXPART-COSMO back trajectories and contributed to the development of the FLEXVAR emission pre-processing. SH contributed to the development of the FLEXPART-COSMO modelling system. PB prepared the paper and figures with

contributions from DB, AS, and JMH. SH, MR, TA, TB, HC, SC, MD, GF, AF, DK, XL, MLe, MLi, MLo, GM, JMW, SOD, BS, MS, PT, GV, and CYK provided atmospheric observation data.

**Competing interests**

The authors declare that they have no conflict of interest.



## Acknowledgements

The VERIFY project has received funding from the European Union's Horizon 2020 research and innovation programme
        under grant agreement No. 776810. We are grateful to ECMWF for providing computing resources under the special
        projects "Improve European and global CH$_4$ and N$_2$O flux inversions (2018 - 2020)" and "Extend and improve CH$_4$ flux
        inversions at global and European scale (2021)". Furthermore, we thank Bradley Matthews for the compilation of
        uncertainties of emissions reported to UNFCCC by EU member states. ICOS Switzerland had been funded by the Swiss
National Science Foundation, in-house contributions and the State Secretariat for Education, Research and Innovation.

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
