# Peer review of "High-resolution inverse modelling of European CH4 emissions using novel FLEXPART-COSMO TM5 4DVAR inverse modelling system"

_Atmospheric Chemistry and Physics, 2022_

## Referee Comment (RC2)

High-resolution inverse modelling of European CH4 emissions using novel FLEXPART-COSMO TM5 4DVAR inverse modelling system

Peter Bergamaschi et. al.

**Overview**

This paper describes an improved, higher resolution, inversion technique, FLEXVAR and compares it to three other related inversion systems. The method is applied to estimating CH4 emissions over Europe. Many sensitivity studies are presented, exploring various setup parameter choices that need to be made. The paper is well written and is very thorough. The presentation of the figures could be improved I believe, they contain too much cramped information leading to a lack of clarity. I have made several scientific points and questions and some minor text corrections.

**Comments**

P2 L42: "of the derived inversion increments" - please clarify what this means

P6 L189: "ignores any error correlation between different observations" – Please briefly discuss the impact of making this assumption between different 3hr periods. It is highly likely that a measurement is strongly correlated with the 3hr measurements either side. How can this effect be minimised? Please explain why this assumption is necessary.

P6 L196 Eq10: Please clarify the equation with the use of brackets especially where items are divided. Please also provide the units of e (I assumed g/m2/s), it would be useful to have this stated.

P10 L301: "The matrix was scaled" – Please be more specific, which matrix? I assumed **B** the error covariance matrix? Fixing this to 20%, how did this compare to the other inversion setups?

P10 L312: "using 25 vertical layers" – I assume these are concentrated near to the surface? Please add a brief sentence describing this selection.

P10 L319: Please describe the rational behind the choice of the temporal correlation time scales.

P10 L321: "function of local emissions" - Are these the prior emissions? What distance is 'local'?

P10 L323 – 325: This last sentence seems out of place to me. How does it relate to the actual "TM5-4DVAR" inversion being discussed in this section?

P12 L369: Please specify which stations are classed as "mountain stations" as some are obvious others are less so e.g. Ochsenkopf, Beromunster etc? Also please describe which stations have the >200m difference between the model and actual orography imposed and what these release heights actually are, maybe simply add extra columns in Table 1 describing this height and class of station.

P12 Tab1: Why are Tacolneston 100m data used? In 2018, the 185m inlet samples much more frequently and is obviously higher and better able to be simulated?

P13 L373: "measurement uncertainty is set to 3 ppb" – How has this been derived? Most if not all observations come with an understanding of this quantity and this can vary between sites and over time. For instance why not use the variability in the CH4 observations across the 3hr period, the data are reported at up to 1 minute resolution? Also there are repeated measurements against standards, the repeatability of these observations also indicate how uncertain the measurements are.

P13 L391: "Natural CH4 emissions were generally used" – When were these not used? The word 'generally' implies that in some instance they were not used, when they are not used, what was used? Table 2 implies they are always used.

P15 L415: "Offshore emissions over the sea are not included in the country totals" – Please explain the impact of this decision? The UK, Netherlands, Norway have significant emissions offshore in the North Sea. In the prior inventories how significant are these, and how does this impact on the conclusions that the UK+Ire totals are similar to what is reported given that the reported totals include these emissions?

P16 L455: Figure 5 is referred to here but is not shown until page 24. I think it should appear earlier in the document.

P20 L530: Maybe a similar comment also could be made about NW France which always appears to have very enhanced emissions but is relatively far from emissions? Or do you think these are real?

P22 L575: "INV-E1-O2 compared to INV-E1-O1" – Please provide the actual values for both simulations as well, along with the +- uncertainties, Figure 5 is too crowded to really extract values.

P24 Fig5: These figures are just too crowded and the different colours are impossible to discern e.g. inv-E1-O1-S2.1 and inv-E1-O1-S1 are indistinguishable. There is just too much information on each plot. On the RHS plots why are the E1 data repeated multiple times? The text for the range lines are blurred onto the lines. Please can this plot be improved?

P25 L606: Please mention that Fig6 resolution has been downgraded compared to earlier, I assume to match TM5?

P25 L613: Please provide the actual emission numbers rather than just the % change.

P27 Fig7: Similar comment to Fig5, it is hard to read the words/numbers in the RH plots, the plots themselves obscure the letters. Please can these be made clearer in some way? Although it is useful to see different inversions compared.

P29 L688: Summarising the results for each country grouping in a table would be very useful here.

P29 L710: "emissions in September (Fig. 5)." - I found this impossible to see as there are too many lines.

**Minor Text Comments**

- P1 L35: add "CH4" emissions
- P2 L55: "effective radiate forcing (ERF)" should be effective radiative forcing
- P2 L56: "preindustrial levels 1750" preindustrial levels in 1750
- P2 L60: "especially on the near-term" especially in the near-term
- P2 L61: "due to the relatively short" due to CH4's relatively short
- P2 L70: "is particularly challenging" consider removing the word "challenging", when compared to CO2
- P3 L86: "which became available" which have become available
- P3 L94: "system is currently developed" system has been developed

- P4 L105: "As alternative" As an alternative
- P4 L106: "applied also the" also applied the
- P4 L111: "which allows to optimize a much" which allows the optimization of a much
- P4 L114: "emissions of individual" emissions from individual
- P5 L155: "allows to optimize emissions" allows the optimization of emissions
- P8 L254: "as function of" insert an "a"
- P10 L306: "which allows to zoom" replace with 'allows the system to zoom'
- P10 L307: "-18°...  $42^{\circ}$ " -18° to  $42^{\circ}$
- P10 L308: "while the global domain" while the remaining global domain
- P13 L375: "as observational base data set" "as the observational base data set"
- P22 L555: "in the country region" "in the prior country region"
- P22 L582: "the emission data" "the prior emission data"
- P24 Fig5 Caption: "and error bars the 2-sigma" "and the error bars are the 2-sigma"
- P25 L601: "Similar as for observation data set O1" "In a similar way, as shown with observation data set O1"
- P25 L608: "of major" "of the major"
- P28 L654: "is only relatively" "is relatively"
- P28 L658: "emission data set E3" "prior emission data set E3"
- P28 L676: "requires to include estimates" "requires the inclusion of estimates"
- P30 L714: "that it allows to constrain a" "that it constrains a"

P30 L718: "emissions 2018 using 24 stations" - "emissions in 2018 using 24 stations"

P30 L725: "derive over large parts of the domain somewhat" – "derive, over large parts of the domain, somewhat"

P30 L733: "in the surroundings of these sites" - "in the vicinity of these sites"

P31 L745: "allow to better reproduce the observations than the" – "allow a better the reproduction of the observations compared to the"

P31 L747: "freedom to optimize" - "freedom used to optimize"

P31 L748: "differences of the inversions, as e.g., the" - " differences in the inversions, e.g., the"

---

## Author Comment (AC1)

**Reply to Anonymous Referee #1**

We thank anonymous referee #1 for the very positive overall evaluation of our manuscript and the very constructive comments.

In the following we repeat the specific comments of the referee in italics and add our replies in regular fonts.

How do the authors plan to use this system in the future? Can the system easily be used to estimate methane emissions over a longer timescale, instead of just 2018? If so, how far back can you go (e.g., availability of observations)? How quickly can it be applied to recent years?

It is planned to use FLEXVAR for EMPA's quasi-operational system to estimate Switzerland's CH4 emissions annually as contribution to the Swiss National Inventory Reporting. Meteorological fields from the COSMO model at horizontal resolution of 7 km × 7 km are available for the years 2002 to 2021. Therefore, the FLEXPART-COSMO back trajectories could be generated for that period with consistent meteorological fields. For analysis periods after 2021, the use of different meteorological input fields could be considered, such as e.g., the COSMO meteo data at horizontal resolution of 1 km × 1 km or analysis (or reanalysis data) from ECMWF IFS. We will add the information about the availability of the 7 km × 7 km COSMO meteo data in the revised manuscript.

Regarding the availability of observations: Various atmospheric data sets are available also for the period before 2018 (our analysis year), however the number of European stations is smaller in previous years. Nevertheless, various analyses have been performed for previous years, e.g., within the VERIFY project for the period 2005-2017.

**Will this system be incorporated in existing "emission verification" efforts?**

See our reply to previous point

The authors have focused their uncertainty analysis on one year. If the framework would be extended to multiple years, it is likely that part of the differences between the inverse set-ups will be constant between years (i.e., systematic), and will therefore not impact a trend analysis. If the authors agree, it would be good to add some discussion on this for context.

We have focussed our analysis on a single year (2018) since the main objective was to demonstrate the use of the new inverse modelling system and to characterize this system in some detail. While the application of FLEXVAR to other years is in principle straightforward (see our reply to first point), the analysis of the uncertainties in derived trends is rather difficult. Indeed, often the argument is made that potential systematic errors of the inverse modelling system (especially of the transport modelling) should be constant and should therefore cancel out when looking on trends. However, meteorological analyses could have e.g., some time-dependent systematic errors (biases) which might be difficult to diagnose. The analysis of trends becomes even more difficult, if the observational data coverage is changing over time.

We agree that the analysis of trends and their uncertainties is an important research question. However, such an analysis is outside the scope of the present paper. In general, the inverse emission estimates suggest some differences between top-down and bottomup, but uncertainties overlap. Given the wide range of sensitivity tests, can the authors provide more insights into the way forward towards reducing the top-down uncertainties such that these differences can be better understood? To what degree can more modeling efforts help, and to what degree do we need a denser observational network? Related to the previous point, will a trend be better constrained than an absolute emission estimate?

In order to better understand the differences between top-down and bottom-up estimates it would be very useful to get independent estimates on smaller regional scales, e.g., using aircraft measurements, and measurement campaigns closer to larger sources, which should help to bridge the gap between top-down and bottom-up estimates.

Of course, further increasing the observational network will improve the top-down estimates. At the same time, however, further improvement of the atmospheric transport models (especially regarding boundary layer height dynamics and vertical transport) as well as improved independent validation (using e.g., 222Rn and boundary layer height measurements) will be essential to improve the inverse modelling and to better characterize their uncertainties.

We will add a short discussion of this in the revised manuscript.

Even if total emissions were much more strongly constrained, then still the significant uncertainty in natural emissions will sustain a large posterior uncertainty. Do the authors see any way to address this challenge from a top-down perspective?

With the current European observational network, it remains very difficult to disentangle natural and anthropogenic sources. In order to better quantify the natural emissions, dedicated measurements closer to natural sources should be performed.

**Other comments:**

I suggest splitting up the lengthy L60-L108 paragraph.

We will split the L60-L108 paragraph.

From Table 3 I understand that the standard spatial correlation is 100km in the FLEXVAR inversions. However, in the FLEXKf and TM5-4DVAR inversions 200km is used (L300 & L319). Why this difference? Could this partly explain why the FLEXVAR inversions reproduce the observations best (i.e., a less stiff state)?

For the FLExKF and TM5-4DVAR inversions we had chosen the default spatial correlation used in these systems. For FLEXVAR, we will investigate the impact of the spatial correlation on the achieved correlation.

L368-374: If I understand correctly, only in-situ observations in the optimal threehourly window are selected, then these are averaged to one daily value per site. I understand this choice, but do the authors consider that any valuable information is lost in averaging out the high-frequency signal, or in the data that are not in this threehour window? Additionally, how is the data from discrete air sampling treated, since there is not the same choice for selecting a time window?

The high-frequency variations of  $CH_4$  mole fractions (e.g., on time scale of minutes) remains very difficult (if not impossible) to simulate with the current 7 km × 7 km COSMO meteo data. Simulation of the high-frequency variations would require much higher spatial and temporal resolution of the meteo data as well as higher resolution of the applied emission inventories.

The given 3-hour time windows were chosen since measurements and model simulations are considered most representative during these 3-hour time windows. Nevertheless, it could be certainly interesting to explore the use of the whole diurnal cycle of the measurements. In this context it is quite encouraging that FLEXVAR simulates in general the diurnal cycles relatively well at most sites (see Figure S7). Nevertheless, the use of the whole diurnal cycle in the inversion would require some more detailed investigations (including the application of appropriate temporal correlations for the observational data).

Discrete air samples were taken depending on their availability (i.e., without application of an additional time window). We will add this information in the revised manuscript.

The authors point out that an advantage of the 4DVAR approach is that, for optimization, the emission grid does not need to be aggregated. However, there is still limited information in the CH4 observations, so that correlation lengths in space and time need to be applied in the optimization. It seems useful to add the effective degrees of freedom that these correlation lengths result in, in addition to the total number of state elements, to compare to the other inverse systems (e.g., near L175).

Clearly the application of spatial and temporal correlation lengths leads to a reduction of the "effective degrees of freedom". However, it is difficult to quantify this. An option could be an analysis of the spectrum of the Eigenvalues (e.g., quantification of number of leading Eigenvectors). However, such an analysis would require substantial additional work and it is not straightforward to apply this to the different inversion techniques (4DVAR vs. Extended Kalman Filter inverse modelling system).

The authors calculate posterior uncertainties in inversions different from the reference inversions. Most importantly, the alternative inversions allow for negative emissions. For this to be a valid strategy, the alternative inversions should converge to similar emissions and reproduce observations similarly as the standard inversions. I could not find any confirmation that this is the case. I understand that qualitatively this approach makes sense and the results seem plausible, but I would like to see some evaluation of this aspect in the manuscript (or supplements).

The additional inversions used to calculate posterior uncertainties (based on the conjugate gradient algorithm) show in general very similar spatial patterns of the inversion increments as the regular inversions (using the semi-lognormal probability density function and the m1qn3 algorithm). Aggregated annual total CH4 emissions agree on average (over all sensitivity inversions) within -1.0% to 2.1% for the 4 country regions discussed in the paper (Germany, France, BENELUX, UK+Ireland) and for individual inversions within -6.4% and 6.5%. We will add this information in the updated version of the manuscript.

Fig. 5 contains a lot of information, and takes a while to fully take in. Having everything differentiated with only color does not help this process. Perhaps it could help to use different markers/linestyles for inversions that start from different priors (e.g., circle for E1, square for E2, triangle for E3)? Then,

statements as in L562-L564 can be more easily seen. Other ways to adjust the colors, or reduce the number of colors, would be desirable: the light-yellow (INV-E1-O1-S3.1) is hardly visible and some of the greens are indistinguishable.

We will update the figure taking into account the suggestions of the reviewer.

I would be interested to see a rough comparison of the computational cost of the different inverse systems. In principle a global TM5-4DVAR inversion is needed for the baseline of the FLEXPART inversions, but this global inversion (as I understand it) only needs to be done once. Once the baseline is determined, are the FLEXPART inversions much faster than the TM5-4DVAR inversions? I expect this to be an important additional advantage of the Lagrangian approach.

Indeed, the FLEXVAR inversions are much faster than a full TM5-4DVAR inversion (with European 1° x 1° zoom), roughly about a factor of 7-10 for the current settings (e.g., including 2 outer loops of the FLEXVAR inversion). However, the computation of the FLEXPART-COSMO back trajectories also requires significant computational resources. Since the FLEXVAR and TM5-4DVAR inversions, however, a quantitative comparison is difficult. Furthermore, the comparison of the total computational costs will depend on the number of FLEXVAR inversions to be performed (since baselines and FLEXPART-COSMO back trajectories need to be computed only once).

---

## Author Comment (AC2)

**Reply to Anonymous Referee #2**

We thank anonymous referee #2 for the very positive overall evaluation of our manuscript and the very constructive comments.

In the following we repeat the specific comments of the referee in italics and add our replies in regular fonts.

*P2 L42: "of the derived inversion increments" – please clarify what this means*

we mean the difference between posterior and prior emissions. This should be a commonly used term. But for clarity we could add a definition in the text (e.g., P16, L452, where this term is used the first time in the main text).

*P6 L189: "ignores any error correlation between different observations" – Please briefly discuss the impact of making this assumption between different 3hr periods. It is highly likely that a measurement is strongly correlated with the 3hr measurements either side. How can this effect be minimised? Please explain why this assumption is necessary.*

We agree that that a strong correlation should be expected between consecutive 3-hourly time windows, e.g., between [12:00 - 15:00] and [15:00 - 18:00]. However, we use only one 3-hourly time window per day (see section 3.1), and the error correlation between the 3-hourly time windows of two consecutive days should be much weaker (and should be largely determined by the actual synoptic situation). For simplicity, we have chosen here the simple assumption of ignoring any error correlation, as assumed also in many previous studies.

*P6 L196 Eq10: Please clarify the equation with the use of brackets especially where items are divided. Please also provide the units of e (I assumed g/m2/s), it would be useful to have this stated.*

We would prefer not to use additional brackets (in order to avoid confusion with the use of brackets for parameters of certain variables), but to use instead a presentation with numerator and denominator to improve the readability of the formula. We will add the units of e.

*P10 L301: "The matrix was scaled" – Please be more specific, which matrix? I assumed B the error covariance matrix? Fixing this to 20%, how did this compare to the other inversion setups?*

Yes, the prior error covariance matrix is meant here. For the FLEXVAR inversions presented in the manuscript the aggregated total uncertainty depends on the corresponding covariance settings. E.g., for 'INV-E1-O1', the aggregated total uncertainty is 12% (1-sigma).

*P10 L312: "using 25 vertical layers" – I assume these are concentrated near to the surface? Please add a brief sentence describing this selection.*

Around 5 layers are within the boundary layer, 10 layers within the free troposphere, and 10 layers in the stratosphere. We will add a short sentence in the revised manuscript.

*P10 L319: Please describe the rational behind the choice of the temporal correlation time scales.*

The rationale behind this choice is that wetlands, rice, and biomass burning have pronounced seasonal cycles, while the "remaining sources" are assumed to have no or only small seasonal variations.

*P10 L321: "function of local emissions" – Are these the prior emissions? What distance is 'local'?*

Local emissions are the emissions of the grid cell in which the corresponding monitoring station is located. We use here the actual emissions (i.e., prior or posterior) of the corresponding model simulation.

*P10 L323 – 325: This last sentence seems out of place to me. How does it relate to the actual "TM5-4DVAR" inversion being discussed in this section?*

The TM5-4DVAR inversions are used both to calculate the baselines for FLEXVAR and for the model comparison.

*P12 L369: Please specify which stations are classed as "mountain stations" as some are obvious others are less so e.g. Ochsenkopf, Beromunster etc? Also please describe which stations have the >200m difference between the model and actual orography imposed and what these release heights actually are, maybe simply add extra columns in Table 1 describing this height and class of station.*

We will add in Table 1 a column indicating which stations are classified as "mountain stations" and a further column with the applied release heights.

*P12 Tab1: Why are Tacolneston 100m data used? In 2018, the 185m inlet samples much more frequently and is obviously higher and better able to be simulated?*

Unfortunately, the 185m sampling height was missing in our list of the FLEXPART simulations. It would have been a significant additional effort to re-run the FLEXPART simulations just for one further station level - therefore we had decided to use for that station just the available 100m level.

Indeed, it would have been preferable to use the data from the 185m sampling height. Nevertheless, the measurements from the 100m level should also be quite representative and well suited for the inverse modelling. Also, the station statistics (comparison of model simulations with measurements) are excellent for the 100m level of this station (see Figure S6).

*P13 L373: "measurement uncertainty is set to 3 ppb" – How has this been derived? Most if not all observations come with an understanding of this quantity and this can vary between sites and over time. For instance why not use the variability in the CH4 observations across the 3hr period, the data are reported at up to 1 minute resolution? Also there are repeated measurements against standards, the repeatability of these observations also indicate how uncertain the measurements are.*

The applied value of 3 ppb is a conservative estimate which should include also potential additional errors (e.g. due to sampling). In any case, however, the modelling errors (model representation

error) are usually much larger - therefore the assumed value for the measurement uncertainty has probably an only minor impact on the inversion results.

*P13 L391: "Natural CH4 emissions were generally used" – When were these not used? The word 'generally' implies that in some instance they were not used, when they are not used, what was used? Table 2 implies they are always used.*

Yes, we always used the natural $CH_4$ emissions from the GCP-CH4 data set. We will delete "generally" in the updated version of the manuscript.

*P15 L415: "Offshore emissions over the sea are not included in the country totals" – Please explain the impact of this decision? The UK, Netherlands, Norway have significant emissions offshore in the North Sea. In the prior inventories how significant are these, and how does this impact on the conclusions that the UK+Ire totals are similar to what is reported given that the reported totals include these emissions?*

Unfortunately, we do not have the information from the gridded emission inventories about the attribution of the offshore emissions to individual countries. However, it is interesting to note that the inversions generally significantly reduce the offshore emissions of the prior inventories (see e.g., Figure 6, where this reduction is clearly visible in the inversions of all three models).

*P16 L455: Figure 5 is referred to here but is not shown until page 24. I think it should appear earlier in the document.*

Figure 5 could be shown indeed earlier, e.g., directly after Figure 1.

*P20 L530: Maybe a similar comment also could be made about NW France which always appears to have very enhanced emissions but is relatively far from emissions? Or do you think these are real?*

(remark: we assume that the referee means here "is relatively far from *observations*")

The derived enhancements over NW France seems to be a much more robust feature of the inversions compared to the enhancements at the eastern domain boundary. The latter depend strongly on the chosen prior uncertainties (Figure S5), while the enhancements over NW France are visible basically in all inversions (and from all three models). However, in the absence of additional studies (e.g., regional measurement campaigns) it remains difficult to judge how realistic these derived emission patterns are. Clearly further independent validation studies will be required to evaluate the quality of the inverse modelling results.

*P22 L575: "INV-E1-O2 compared to INV-E1-O1" – Please provide the actual values for both simulations as well, along with the +- uncertainties, Figure 5 is too crowded to really extract values.*

We will add the actual values and their uncertainties in the updated version of the manuscript.

*P24 Fig5: These figures are just too crowded and the different colours are impossible to discern e.g. inv-E1-O1-S2.1 and inv-E1-O1-S1 are indistinguishable. There is just too much information on each plot. On the RHS plots why are the E1 data repeated multiple times? The text for the range lines are blurred onto the lines. Please can this plot be improved?*

We will update the Fig. 5. However, we plan to keep the data of the prior emissions for each inversion. Even if the prior values are identical for a given prior inventory (E1, E2, E3), their uncertainty depends on the chosen model covariance settings.

*P25 L606: Please mention that Fig6 resolution has been downgraded compared to earlier, I assume to match TM5?*

No, the resolution has not been downgraded. For the model comparison shown in this figure we used E3 as prior inventory, which had been provided at horizontal resolution of 1° × 1°.

*P25 L613: Please provide the actual emission numbers rather than just the % change.*

We will add the actual emission numbers in the revised version of the manuscript.

*P27 Fig7: Similar comment to Fig5, it is hard to read the words/numbers in the RH plots, the plots themselves obscure the letters. Please can these be made clearer in some way? Although it is useful to see different inversions compared.*

We will update also the Fig. 7 (consistently with the planned update of Fig. 5).

*P29 L688: Summarising the results for each country grouping in a table would be very useful here.*

We will consider to add a table in the supplementary material.

*P29 L710: "emissions in September (Fig. 5)." - I found this impossible to see as there are too many lines.*

We will try to improve the visibility of the seasonal variation of the prior emissions

*Minor Text Comments*

We will directly adopt most of the suggested minor text comments.

---

## Author Response (AR1)

**Reply to Anonymous Referee #1**

In the following we repeat the comments of the referee in italics (and black) and add our replies in blue and regular fonts.

*Bergamaschi et al. present a novel inverse modeling framework for estimating European methane emissions. The analysis includes a detailed investigation of the sensitivities and uncertainties in the inverse system, and in general the estimated emissions are shown to be robust. The authors have gone through the additional, considerable effort of performing some of the inversions in different inverse systems, to highlight the advantage of their new approach. The analysis is comprehensive, and addresses all important aspects that the presentation of a new system should address. The system itself is a valuable addition for estimating methane emissions in a time where, as the authors point out, methane has become an important mitigation target.*

We thank anonymous referee #1 for the very positive overall evaluation of our manuscript and the very constructive comments.

*My most important comment is that while the technical aspects of the analysis are strongly highlighted and generally well-explained, I miss reflection on the context of the inverse system especially in later parts of the manuscript (discussion or conclusion). Therefore, I think it is important to address some additional relevant questions in the manuscript, such as:*

We have addressed the "additional relevant questions" as detailed in the following replies to the specific points raised by referee #1.

*How do the authors plan to use this system in the future? Can the system easily be used to estimate methane emissions over a longer timescale, instead of just 2018? If so, how far back can you go (e.g., availability of observations)? How quickly can it be applied to recent years?*

It is planned to use FLEXVAR for EMPA's quasi-operational system to estimate Switzerland's $CH_4$ emissions annually as contribution to the Swiss National Inventory Reporting. This planned use of FLEXVAR is now mentioned at the end of the Conclusions.

Meteorological fields from the COSMO model at horizontal resolution of 7 km × 7 km are available for the years 2002 to 2021. Therefore, the FLEXPART-COSMO back trajectories could be generated for that period with consistent meteorological fields. For analysis periods after 2021, the use of different meteorological input fields could be considered, such as e.g., the meteorological data from the ECMWF IFS model at high resolution (0.1° x 0.1°) or the operational COSMO meteorological data from MeteoSwiss at horizontal resolution of 1 km × 1 km (which, however, cover only a smaller part of the European domain). We have added the information about the availability of the 7 km × 7 km COSMO meteorological data and potential alternative data sets for analysis periods after 2021 at the end of the Conclusions.

Regarding the availability of observations: Various atmospheric data sets are available also for the period before 2018 (our analysis year), however the number of European stations is smaller in previous years. Nevertheless, various analyses have been performed for previous years, e.g., within the VERIFY project for the period 2005-2017.

*Will this system be incorporated in existing "emission verification" efforts?*

The planned use of FLEXVAR for "emission verification efforts" is now mentioned at the end of the Conclusions (see also our reply to previous point).

*The authors have focused their uncertainty analysis on one year. If the framework would be extended to multiple years, it is likely that part of the differences between the inverse set-ups will be constant between years (i.e., systematic), and will therefore not impact a trend analysis. If the authors agree, it would be good to add some discussion on this for context.*

We have focussed our analysis on a single year (2018) since the main objective was to demonstrate the use of the new inverse modelling system and to characterize this system in some detail. While the application of FLEXVAR to other years is in principle straightforward (see our reply to first point), the analysis of the uncertainties in derived trends is rather difficult. Indeed, often the argument is made that potential systematic errors of the inverse modelling system (especially of the transport modelling) should be constant and should therefore cancel out when looking on trends. However, meteorological analyses could have e.g., some time-dependent systematic errors (biases) which might be difficult to diagnose. The analysis of trends becomes even more difficult, if the observational data coverage is changing over time.

We agree that the analysis of trends and their uncertainties is an important research question. However, such an analysis is outside the scope of the present paper.

*In general, the inverse emission estimates suggest some differences between top-down and bottom-up, but uncertainties overlap. Given the wide range of sensitivity tests, can the authors provide more insights into the way forward towards reducing the top-down uncertainties such that these differences can be better understood? To what degree can more modeling efforts help, and to what degree do we need a denser observational network? Related to the previous point, will a trend be better constrained than an absolute emission estimate?*

A first step should be further studies to assess independently the quality of the top-down estimates. Such assessments should include the comparison with further inverse models, comparison with independent regional emission estimates (e.g., based on aircraft or satellite measurements), and a more detailed validation of the applied atmospheric transport models (using e.g., $^{222}$Rn and boundary layer height measurements). We have added this at the end of the conclusions.

Of course, further increasing the observational network is important to reduce the uncertainties in the top-down estimates. At the same time, however, further improvement of the atmospheric transport models (especially regarding boundary layer height dynamics and vertical transport) will be essential to improve the inverse modelling. Regarding the question about trend, see our reply to previous point.

*Even if total emissions were much more strongly constrained, then still the significant uncertainty in natural emissions will sustain a large posterior uncertainty. Do the authors see any way to address this challenge from a top-down perspective?*

With the current European observational network, it remains very difficult to disentangle natural and anthropogenic sources. In order to better quantify the natural emissions, dedicated measurements closer to natural sources should be performed.

***Other comments:***

*I suggest splitting up the lengthy L60-L108 paragraph.*

We have split the L60-L108 paragraph into several smaller paragraphs.

*From Table 3 I understand that the standard spatial correlation is 100km in the FLEXVAR inversions. However, in the FLEXKf and TM5-4DVAR inversions 200km is used (L300 & L319). Why this difference? Could this partly explain why the FLEXVAR inversions reproduce the observations best (i.e., a less stiff state)?*

For the FLExKF and TM5-4DVAR inversions we had chosen the default spatial correlation used in these systems. For FLEXVAR, we have further investigated the impact of the covariance settings on the achieved correlation. Increasing the correlation length from 100 km to 200 km (INV-E1-O1-S3.2 vs. INV-E1-O1) is indeed slightly deteriorating the statistical performance (comparison of model simulations and observations), but nevertheless the statistical performance remains better compared to FLExKF. On the other hand, FLExKF applies a higher prior uncertainty than FLEXVAR in the model comparison discussed in the paper. For FLEXVAR, increasing the prior uncertainty from 100% to 200% (INV-E1-O1-S4.2 vs. INV-E1-O1), is slightly improving the statistical performance, i.e., partly compensating the effect of a larger correlation length.

We have now included in section 4.3 and in the Conclusions a short remark that also the different covariance settings may partly contribute to the difference in the statistical performance of FLExKF vs. FLEXVAR.

*L368-374: If I understand correctly, only in-situ observations in the optimal threehourly window are selected, then these are averaged to one daily value per site. I understand this choice, but do the authors consider that any valuable information is lost in averaging out the high-frequency signal, or in the data that are not in this threehour window? Additionally, how is the data from discrete air sampling treated, since there is not the same choice for selecting a time window?*

The high-frequency variations of $CH_4$ mole fractions (e.g., on time scale of minutes) remains very difficult (if not impossible) to simulate with the current 7 km $\times$ 7 km COSMO meteorological data. Simulation of the high-frequency variations would require much higher spatial and temporal resolution of the meteorological data as well as higher resolution of the applied emission inventories. Furthermore, the use of high-frequency data may be hampered by temporally correlated errors (especially in model transport).

The given 3-hour time windows were chosen since measurements and model simulations are considered most representative during these 3-hour time windows. During the early afternoon the boundary layers is usually fully developed, the boundary layer height relatively constant, and the tracers well mixed within the boundary layer. Therefore, for surface stations the early afternoon data are assimilated. For mountain sites, on the other hand, the night-time data are considered most representative, when the station is usually within the free troposphere.

Nevertheless, it could be certainly interesting to explore the use of the whole diurnal cycle of the measurements. In this context it is quite encouraging that FLEXVAR simulates in general the diurnal cycles relatively well at most sites (see Figure S7). However, the use of the whole diurnal cycle in the inversion would require some more detailed investigations (including the application of appropriate temporal correlations for the observational data).

Discrete air samples were taken as available, i.e., without any temporal selection. We have added this information in section 3.1.

*The authors point out that an advantage of the 4DVAR approach is that, for optimization, the emission grid does not need to be aggregated. However, there is still limited information in the CH4 observations, so that correlation lengths in space and time need to be applied in the optimization. It seems useful to add the effective degrees of freedom that these correlation lengths result in, in addition to the total number of state elements, to compare to the other inverse systems (e.g., near L175).*

Clearly the application of spatial and temporal correlation lengths leads to a reduction of the "effective degrees of freedom". However, it is difficult to quantify this. An option could be an analysis of the spectrum of the Eigenvalues (e.g., quantification of number of leading Eigenvectors). However, such an analysis would require substantial additional work and it is not straightforward to apply this in a consistent way to the different inversion techniques (4DVAR vs. Extended Kalman Filter inverse modelling system).

*The authors calculate posterior uncertainties in inversions different from the reference inversions. Most importantly, the alternative inversions allow for negative emissions. For this to be a valid strategy, the alternative inversions should converge to similar emissions and reproduce observations similarly as the standard inversions. I could not find any confirmation that this is the case. I understand that qualitatively this approach makes sense and the results seem plausible, but I would like to see some evaluation of this aspect in the manuscript (or supplements).*

The additional inversions used to calculate posterior uncertainties (based on the conjugate gradient algorithm) show in general very similar spatial patterns of the inversion increments as the regular inversions (using the semi-lognormal probability density function and the m1qn3 algorithm). Aggregated annual total $CH_4$ emissions agree on average (over all sensitivity inversions) within -1.0% to 2.1% for the 4 country regions discussed in the paper (Germany, France, BENELUX, UK+Ireland) and for individual inversions within -6.4% and 6.5%. We have added this information in section 2.2.1.

*Fig. 5 contains a lot of information, and takes a while to fully take in. Having everything differentiated with only color does not help this process. Perhaps it could help to use different markers/linestyles for inversions that start from different priors (e.g., circle for E1, square for E2, triangle for E3)? Then, statements as in L562-L564 can be more easily seen. Other ways to adjust the colors, or reduce the number of colors, would be desirable: the light-yellow (INV-E1-O1-S3.1) is hardly visible and some of the greens are indistinguishable.*

We have updated the figure (please note: following the suggestion of reviewer #2 we have moved the figure - therefore in the revised version the previous Fig.5 is now Fig.2). As suggested by the reviewer, we use now different symbols for different priors (as well as for the corresponding posterior emissions) in the right panel of the figure. Also, some colours have been updated to make them more clearly distinguishable (or better visible). Furthermore, the size of the labels (indicating the different inversions) has been enlarged in the right panel (as well as the width of the right panel).

*I would be interested to see a rough comparison of the computational cost of the different inverse systems. In principle a global TM5-4DVAR inversion is needed for the baseline of the FLEXPART inversions, but this global inversion (as I understand it) only needs to be done once. Once the baseline is determined, are the FLEXPART inversions much faster than the TM5-4DVAR inversions? I expect this to be an important additional advantage of the Lagrangian approach.*

Indeed, the FLEXVAR inversions are much faster than a full TM5-4DVAR inversion (with European 1° x 1° zoom), roughly about a factor of 7-10 for the current settings (e.g., including 2 outer loops of the FLEXVAR inversion). However, the computation of the FLEXPART-COSMO back trajectories also requires significant computational resources. Since the FLEXPART-COSMO back trajectories were computed on a different computing platform than the FLEXVAR and TM5-4DVAR inversions, however, a quantitative comparison is difficult. Furthermore, the comparison of the total computational costs will depend on the number of FLEXVAR inversions to be performed (since baselines and FLEXPART-COSMO back trajectories need to be computed only once).

**Reply to Anonymous Referee #2**

In the following we repeat the comments of the referee in italics (and black) and add our replies in blue and regular fonts.

*This paper describes an improved, higher resolution, inversion technique, FLEXVAR and compares it to three other related inversion systems. The method is applied to estimating CH4 emissions over Europe. Many sensitivity studies are presented, exploring various setup parameter choices that need to be made. The paper is well written and is very thorough.*

We thank anonymous referee #2 for the very positive overall evaluation of our manuscript and the very constructive comments.

*The presentation of the figures could be improved I believe, they contain too much cramped information leading to a lack of clarity. I have made several scientific points and questions and some minor text corrections.*

We have addressed the "scientific points and questions" (including an update of the figures) and the minor text corrections as detailed in the following replies to the specific points raised by referee #2.

*P2 L42: "of the derived inversion increments" – please clarify what this means*

"inversion increments" mean the difference between posterior and prior emissions. This should be a commonly used term. For clarity we have added a definition in the text at the beginning of section 4.1.1, where this term is used the first time in the main text.

*P6 L189: "ignores any error correlation between different observations" – Please briefly discuss the impact of making this assumption between different 3hr periods. It is highly likely that a measurement is strongly correlated with the 3hr measurements either side. How can this effect be minimised? Please explain why this assumption is necessary.*

We agree that that a strong correlation should be expected between consecutive 3-hourly time windows, e.g., between [12:00 - 15:00] and [15:00 - 18:00]. However, we use only one 3-hourly time window per day (see section 3.1), and the error correlation between the 3-hourly time windows of two consecutive days should be much weaker (and should be largely determined by the actual synoptic situation). For simplicity, we have chosen here the simple assumption of ignoring any error correlation, as assumed also in many previous studies.

*P6 L196 Eq10: Please clarify the equation with the use of brackets especially where items are divided. Please also provide the units of e (I assumed g/m2/s), it would be useful to have this stated.*

We have updated Eq10 (however not using additional brackets, but using instead a presentation with numerator and denominator) to improve the readability of the formula. We have added a short sentence explaining the meaning of $e_{i,j,t}(x_{i,j,t})$ and its units.

*P10 L301: "The matrix was scaled" – Please be more specific, which matrix? I assumed B the error covariance matrix? Fixing this to 20%, how did this compare to the other inversion setups?*

Yes, the prior error covariance matrix is meant here. This has been added now to the text. For the FLEXVAR inversions presented in the manuscript the aggregated total uncertainty depends on the corresponding covariance settings. E.g., for 'INV-E1-O1', the aggregated total uncertainty is 12% (1-sigma). The different prior uncertainties per country are visible in Fig. 7 (however sometimes exceeding the scale) and are compiled in the new Table S4.

*P10 L312: "using 25 vertical layers" – I assume these are concentrated near to the surface? Please add a brief sentence describing this selection.*

About 5 layers represent the boundary layer, 10 layers the free troposphere, and 10 layers the stratosphere. This has been added now in the manuscript.

*P10 L319: Please describe the rational behind the choice of the temporal correlation time scales.*

The rationale behind this choice is that wetlands, rice, and biomass burning have pronounced seasonal cycles, while the "remaining sources" are assumed to have no or only small seasonal variations. This has now been added in the text.

*P10 L321: "function of local emissions" – Are these the prior emissions? What distance is 'local'?*

Local emissions are the emissions of the grid cell in which the corresponding monitoring station is located. This has now been added in the text. We use here the actual emissions (i.e., prior or posterior) of the corresponding model simulation.

*P10 L323 – 325: This last sentence seems out of place to me. How does it relate to the actual "TM5-4DVAR" inversion being discussed in this section?*

The TM5-4DVAR inversions are used both for the model comparison and to calculate the baselines for FLEXVAR. "This last sentence" refers to the latter, which should be clear as the sentence is introduced with "For the coupling of the FLEXPART-COSMO inversions..."

*P12 L369: Please specify which stations are classed as "mountain stations" as some are obvious others are less so e.g. Ochsenkopf, Beromunster etc? Also please describe which*

*stations have the >200m difference between the model and actual orography imposed and what these release heights actually are, maybe simply add extra columns in Table 1 describing this height and class of station.*

We have added in Table 1 a column with the applied release heights (for stations where the difference between model and actual orography is greater than 200m, the release heights are listed as heights above sea level ("asl"); otherwise, the release height above model surface ("agl") is given. In addition, we have added a further column ("M") which indicates the stations which are classified as "mountain stations".

*P12 Tab1: Why are Tacolneston 100m data used? In 2018, the 185m inlet samples much more frequently and is obviously higher and better able to be simulated?*

Unfortunately, the 185m sampling height was missing in our list of the FLEXPART simulations. It would have been a significant additional effort to re-run the FLEXPART simulations just for one further station level - therefore we had decided to use for that station just the available 100m level.

Indeed, it would have been preferable to use the data from the 185m sampling height. Nevertheless, the measurements from the 100m level should also be quite representative and well suited for the inverse modelling. Also, the station statistics (comparison of model simulations with measurements) are excellent for the 100m level of this station (see Figure S6). Furthermore, we note that e.g., *Andrews et al.* [2014] (see their Section 7.6) concluded that data from ~100m above ground level are generally sufficient for most applications with current models.

Andrews, A. E., Kofler, J. D., Trudeau, M. E., Williams, J. C., Neff, D. H., Masarie, K. A., Chao, D. Y., Kitzis, D. R., Novelli, P. C., Zhao, C. L., Dlugokencky, E. J., Lang, P. M., Crotwell, M. J., Fischer, M. L., Parker, M. J., Lee, J. T., Baumann, D. D., Desai, A. R., Stanier, C. O., De Wekker, S. F. J., Wolfe, D. E., Munger, J. W., and Tans, P. P.: $CO_2$, CO, and $CH_4$ measurements from tall towers in the NOAA Earth System Research Laboratory's Global Greenhouse Gas Reference Network: instrumentation, uncertainty analysis, and recommendations for future high-accuracy greenhouse gas monitoring efforts, Atmos. Meas. Tech., 7, 647–687, https://doi.org/10.5194/amt-7-647-2014, 2014.

*P13 L373: "measurement uncertainty is set to 3 ppb" – How has this been derived? Most if not all observations come with an understanding of this quantity and this can vary between sites and over time. For instance why not use the variability in the CH4 observations across the 3hr period, the data are reported at up to 1 minute resolution? Also there are repeated measurements against standards, the repeatability of these observations also indicate how uncertain the measurements are.*

The applied value of 3 ppb is a conservative estimate which should include also potential additional errors (e.g., due to sampling). In any case, however, the modelling errors (model representation error) are usually much larger - therefore the assumed value for the measurement uncertainty has probably an only minor impact on the inversion results.

*P13 L391: "Natural CH4 emissions were generally used" – When were these not used? The word 'generally' implies that in some instance they were not used, when they are not used, what was used? Table 2 implies they are always used.*

Yes, we always used the natural CH$_4$ emissions from the GCP-CH4 data set. We have deleted "generally" in the updated version of the manuscript.

*P15 L415: "Offshore emissions over the sea are not included in the country totals" – Please explain the impact of this decision? The UK, Netherlands, Norway have significant emissions offshore in the North Sea. In the prior inventories how significant are these, and how does this impact on the conclusions that the UK+Ire totals are similar to what is reported given that the reported totals include these emissions?*

Unfortunately, we do not have the information from the gridded emission inventories about the attribution of the offshore emissions to individual countries. However, it is interesting to note that the inversions generally significantly reduce the offshore emissions of the prior inventories (see e.g., Figure 6, where this reduction is clearly visible in the inversions of all three models). This is now also briefly mentioned in section 4.3.

*P16 L455: Figure 5 is referred to here but is not shown until page 24. I think it should appear earlier in the document.*

We have moved Figure 5. Now it is shown directly after Figure 1. Therefore, the figure numbers have been updated as follows:

Figure 5 -> Figure 2

Figure 2 -> Figure 3

Figure 3 -> Figure 4

Figure 4 -> Figure 5

*P20 L530: Maybe a similar comment also could be made about NW France which always appears to have very enhanced emissions but is relatively far from emissions? Or do you think these are real?*

(remark: we assume that the referee means here "is relatively far from *observations*")

The derived enhancements over NW France seems to be a much more robust feature of the inversions compared to the enhancements at the eastern domain boundary. The latter depend strongly on the chosen prior uncertainties (Figure S5), while the enhancements over NW France are visible basically in all inversions (and from all three models). However, in the absence of additional studies (e.g., regional measurement campaigns) it remains difficult to judge how realistic these derived emission patterns are. Clearly further independent validation studies will be required to evaluate the quality of the inverse modelling results.

*P22 L575: "INV-E1-O2 compared to INV-E1-O1" – Please provide the actual values for both simulations as well, along with the +- uncertainties, Figure 5 is too crowded to really extract values.*

We have added the actual values. Their uncertainties were already provided in the following sentence ("The significant additional observational constraints for UK+Ireland are also reflected in the significantly lower posterior uncertainty for INV-E1-O2 (2-sigma uncertainty: 0.6 Tg $CH_4$ $yr^{-1}$) compared to INV-E1-O1 (2-sigma uncertainty: 1.6 Tg $CH_4$ $yr^{-1}$; Fig. 2)".

*P24 Fig5: These figures are just too crowded and the different colours are impossible to discern e.g. inv-E1-O1-S2.1 and inv-E1-O1-S1 are indistinguishable. There is just too much information on each plot. On the RHS plots why are the E1 data repeated multiple times? The text for the range lines are blurred onto the lines. Please can this plot be improved?*

We have updated the figure. Some colours have been changed to make them more clearly distinguishable. We have re-arranged the plotting of the data, putting now in the right panel of the figure the annual total prior and posterior data of each inversion into a single slot (separated by light grey lines). Furthermore, we have increased the distance of the labels from the symbols (to avoid the overlap of the labels with the symbols / error bars) and increased the size of the labels.

We have kept the prior data for each individual inversion, since their uncertainties are variable (depending on the chosen model covariance settings). However, following the suggestion of reviewer #1, we use now different symbols for the different priors (and the related posterior data).

*P25 L606: Please mention that Fig6 resolution has been downgraded compared to earlier, I assume to match TM5?*

No, the resolution has not been downgraded. For the model comparison shown in this figure we used E3 as prior inventory, which had been provided at horizontal resolution of $1° \times 1°$.

*P25 L613: Please provide the actual emission numbers rather than just the % change.*

We have added the actual emission numbers.

*P27 Fig7: Similar comment to Fig5, it is hard to read the words/numbers in the RH plots, the plots themselves obscure the letters. Please can these be made clearer in some way? Although it is useful to see different inversions compared.*

We have updated also the Fig. 7, consistently with the update of (previous) Fig. 5 (now Fig. 2).

*P29 L688: Summarising the results for each country grouping in a table would be very useful here.*

We have added a table (new table S4 in the supplementary material) compiling the results for all inversions for all country groups.

*P29 L710: "emissions in September (Fig. 5)." - I found this impossible to see as there are too many lines.*

We have put the seasonal variations of the prior emissions now to the foreground. Therefore, they are now clearly visible for all three emission data sets (E1, E2, E3).

***Minor Text Comments***

*P1 L35: add "CH4" emissions*
has been corrected

*P2 L55: "effective radiate forcing (ERF)" – should be effective radiative forcing*
has been corrected

*P2 L56: "preindustrial levels 1750" - preindustrial levels in 1750*
has been corrected

*P2 L60: "especially on the near-term" - especially in the near-term*
has been corrected

*P2 L61: "due to the relatively short" - due to CH4's relatively short*
has been corrected

*P2 L70: "is particularly challenging" – consider removing the word "challenging", when compared to CO2*
We prefer to keep the work "challenging". E.g., $CO_2$ emissions from fossil fuels can be quantified much more accurately by bottom-up techniques compared to almost all $CH_4$ sources, mainly because the emission factors of $CO_2$ fossil fuels sources are rather well known (while emission factors of most $CH_4$ sources are rather variable). Nevertheless, we agree that estimating $CO_2$ sources and sinks is also quite challenging (especially estimates of natural $CO_2$ fluxes).

*P3 L86: "which became available" – which have become available*
has been corrected

*P3 L94: "system is currently developed" - system has been developed*
has been corrected

*P4 L105: "As alternative" - As an alternative*
has been corrected

*P4 L106: "applied also the" – also applied the*
has been corrected

*P4 L111: "which allows to optimize a much" - which allows the optimization of a much*
has been corrected

*P4 L114: "emissions of individual" - emissions from individual*
has been corrected

*P5 L155: "allows to optimize emissions" – allows the optimization of emissions*
has been corrected

*P8 L254: "as function of" – insert an "a"*
has been corrected

*P10 L306: "which allows to zoom" – replace with 'allows the system to zoom'*
has been corrected

*P10 L307: "-18°... 42°" – -18° to 42°*
has been corrected

*P10 L308: "while the global domain" – while the remaining global domain*
has been corrected

*P13 L375: "as observational base data set" – "as the observational base data set"*
has been corrected

*P22 L555: "in the country region" – "in the prior country region"*
has been corrected (assuming that this refers to P22 L**559**)

*P22 L582: "the emission data" – "the prior emission data"*
has been corrected

*P24 Fig5 Caption: "and error bars the 2-sigma" – "and the error bars are the 2-sigma"*
has been corrected

*P25 L601: "Similar as for observation data set O1" – "In a similar way, as shown with observation data set O1"*
has been corrected

*P25 L608: "of major" – "of the major"*
has been corrected

*P28 L654: "is only relatively" – "is relatively"*
has been corrected

*P28 L658: "emission data set E3" – "prior emission data set E3"*
has been corrected

*P28 L676: "requires to include estimates" – "requires the inclusion of estimates"*
has been corrected

*P30 L714: "that it allows to constrain a" – "that it constrains a"*
we prefer to keep the original wording, since the proposed change is changing the statement
(in fact 4DVAR techniques can also be used for systems with a smaller number of variables)

*P30 L718: "emissions 2018 using 24 stations" – "emissions in 2018 using 24 stations"*
has been corrected

*P30 L725: "derive over large parts of the domain somewhat" – "derive, over large parts of the domain, somewhat"*
has been corrected

*P30 L733: "in the surroundings of these sites" – "in the vicinity of these sites"*
has been corrected

*P31 L745: "allow to better reproduce the observations than the" – "allow a better the reproduction of the observations compared to the"*
has been corrected (but leaving out the suggested "the" before "reproduction")

*P31 L747: "freedom to optimize" – "freedom used to optimize"*
we prefer to keep the original wording

*P31 L748: "differences of the inversions, as e.g., the" – " differences in the inversions, e.g., the*
has been corrected

---

## Referee Report (RR1)

I appreciate the authors taking the time to address my previous comments. I had, however, hoped that the authors would have been a bit more creative in addressing some of my comments, which would not have required additional analysis. Here, I provide small suggestions on further addressing some comments that I think deserve more attention.

- For a new system aimed at emission verification, it seems strange to not in any way mention trends in the manuscript. Emission targets are generally set in terms of relative emission reductions, rather than absolute emissions, so it seems that that would be where verification efforts are focused. I would like the authors to reconsider adding a few sentences on the practicalities of moving to a longer timeseries, and on possible implications on the results. This is partly done in the new L765-774 paragraph, but this is for some reason focused on Switzerland, whereas the presented study is for Europe. For a reader who is not completely up-to-date on existing verification efforts, it is hard to otherwise place this study in context.

- L672-674: This is a very implicit way of stating that the different model systems use different spatial correlations. I suggest making it more explicit, since it seems important.
  In addition, an easy way to check the influence of the different spatial correlation lengths on this comparison would be to have a look at the same observational error statistics for inversion INV-E1-O1-S3.2. Have the authors done this? Is FLEXVAR-200km still better performing than FLExKF?
  I understand that the difference in performance is relatively small, but the authors themselves raise the point, so I would like them to make these small efforts to find more clearly where the difference comes from.

- The authors go through considerable effort to compare different inverse modeling systems. I understand that it is hard to compare explicitly and quantitatively the computational costs of the different systems. However, I would like to see some small discussion of the practical (dis)advantages of the different inverse systems (as in the author's reply to the final point of my first review), since this is an important part of choosing which inverse system to use.

---

## Author Response (AR2)

**Reply to 2nd review of Anonymous Referee #1**

In the following we repeat the comments of the referee in italics (and black) and add our replies in blue and regular fonts.

*I appreciate the authors taking the time to address my previous comments. I had, however, hoped that the authors would have been a bit more creative in addressing some of my comments, which would not have required additional analysis. Here, I provide small suggestions on further addressing some comments that I think deserve more attention.*

We thank anonymous referee #1 for his/her 2nd review and reply to the specific 3 points below.

*For a new system aimed at emission verification, it seems strange to not in any way mention trends in the manuscript. Emission targets are generally set in terms of relative emission reductions, rather than absolute emissions, so it seems that that would be where verification efforts are focused. I would like the authors to reconsider adding a few sentences on the practicalities of moving to a longer timeseries, and on possible implications on the results. This is partly done in the new L765-774 paragraph, but this is for some reason focused on Switzerland, whereas the presented study is for Europe. For a reader who is not completely up-to-date on existing verification efforts, it is hard to otherwise place this study in context.*

We had added in the previous revised version the requested information about the practicalities to apply FLEXVAR to longer time series as follows:

"FLEXVAR inversions with the configuration presented in this paper could be performed for the years 2002 to 2021, the period for which meteorological fields from the COSMO-7 model at 7 km × 7 km resolution are available. For analysis periods after 2021, the use of different high-resolution meteorological input fields could be considered, such as e.g., the operational analysis data from the ECMWF IFS model at high resolution (0.1° x 0.1°) or the operational MeteoSwiss COSMO-1 analysis at horizontal resolution of 1 km × 1 km. COSMO-1, however, is limited to the larger Alpine area, but can be nested into FLEXPART-IFS. A FLEXPART-COSMO modelling system using COSMO-1 has already been developed by Empa, including a modification of the turbulence parameterization [Katharopoulos et al., 2022], which is required owing to the very high resolution of 1 km × 1 km."

Only the COSMO-1 data are limited to the larger Alpine area, while the ECMWF IFS model at high resolution (0.1° x 0.1°) are available even globally. Therefore, potential future applications of FLEXVAR for other time periods are certainly not limited to Switzerland. With the existing COSMO-7 data, FLEXVAR inversions could be directly performed for the years 2002 to 2021 (on the COSMO-7 domain as presented in the current paper for 2018).

As stated in our previous reply, however, the analysis of emission trends is beyond the scope of the present paper. We emphasize again, that in particular the analysis of the uncertainties in derived emission trends is rather challenging.

*L672-674: This is a very implicit way of stating that the different model systems use different spatial correlations. I suggest making it more explicit, since it seems important.*

*In addition, an easy way to check the influence of the different spatial correlation lengths on this comparison would be to have a look at the same observational error statistics for inversion INV-E1-O1-S3.2. Have the authors done this? Is FLEXVAR-200km still better performing than FLExKF?*

*I understand that the difference in performance is relatively small, but the authors themselves raise the point, so I would like them to make these small efforts to find more clearly where the difference comes from.*

As stated in our previous reply we had further investigated the impact of the covariance settings on the achieved correlation, including the analysis of inversion INV-E1-O1-S3.2 (increasing the correlation length from 100 km to 200 km) and INV-E1-O1-S4.2 (increasing the prior uncertainty from 100% to 200%). However, these sensitivity inversions have been performed using emission data set E1 and observation data set O1 and should therefore be compared with the corresponding FLEXVAR and FLExKF inversions (i.e., FLEXVAR INV-E1-O1 and FLExKF E1-O1).

Following the request of the reviewer, we have added now a short summary of this additional analysis also the revised manuscript:

"E.g., the FLEXVAR inversion INV-E3-O2 used for the model comparison applies a smaller spatial correlation length ($L_{corr}$ = 100 km) compared to FLExKF ($L_{corr}$ = 200 km). Comparison of FLEXVAR inversions INV-E1-O1 and INV-E1-O1-S3.2 shows that increasing the correlation length from 100 km to 200 km is indeed slightly deteriorating the statistical performance (mean correlation coefficient and mean rms difference), but nevertheless FLEXVAR (INV-E1-O1-S3.2) still performs slightly better compared to FLExKF (inversion FLExKF E1-O1). On the other hand, FLExKF applies a higher prior uncertainty than FLEXVAR (Table S4) in the model comparison discussed in the paper. For FLEXVAR, increasing the prior uncertainty from 100% to 200% (INV-E1-O1-S4.2 vs. INV-E1-O1), is slightly improving the statistical performance, i.e., partly compensating the effect of a larger correlation length (results not shown)."

*The authors go through considerable effort to compare different inverse modeling systems. I understand that it is hard to compare explicitly and quantitatively the computational costs of the different systems. However, I would like to see some small discussion of the practical (dis)advantages of the different inverse systems (as in the author's reply to the final point of my first review), since this is an important part of choosing which inverse system to use.*

As stated in our previous reply, the FLEXPART-COSMO back trajectories were computed on a different computing platform than the FLEXVAR and TM5-4DVAR inversions. Therefore, we cannot compare quantitatively the required computational resources. Moreover, such a comparison strongly depends on the specific application, e.g., number of stations used (e.g., computational costs for FLEXPART back trajectories scale directly with number of stations, while the number of stations has only a minor impact on the costs of TM5-4DVAR inversions) and number of FLEXVAR inversions to be performed (since FLEXPART-COSMO back trajectories and TM5 baselines need to be computed only once). Apart from

this, it is not clear, what exactly the reviewer means by "practical (dis)advantages of the different inverse systems"